# The guide sRNA sequence determines the activity level of box C/D RNPs

**Andrea Graziadei[1,2], Frank Gabel[3,4], John Kirkpatrick[2,5], Teresa Carlomagno[2,5]\***

[1]European Molecular Biology Laboratory, Structural and Computational Biology , Heidelberg, Germany; [2]Leibniz University Hannover, Centre for Biomolecular Drug Research, Hannover, Germany; [3]University Grenoble Alpes, CEA, CNRS IBS, Grenoble, France; [4]Institut Laue-Langevin, Grenoble, France; [5]Helmholtz Centre for Infection Research, Group of Structural Chemistry, Braunschweig, Germany

**Abstract** 2'-O-rRNA methylation, which is essential in eukaryotes and archaea, is catalysed by the Box C/D RNP complex in an RNA-guided manner. Despite the conservation of the methylation sites, the abundance of site-specific modifications shows variability across species and tissues, suggesting that rRNA methylation may provide a means of controlling gene expression. As all Box C/D RNPs are thought to adopt a similar structure, it remains unclear how the methylation efficiency is regulated. Here, we provide the first structural evidence that, in the context of the Box C/D RNP, the affinity of the catalytic module fibrillarin for the substrate–guide helix is dependent on the RNA sequence outside the methylation site, thus providing a mechanism by which both the substrate and guide RNA sequences determine the degree of methylation. To reach this result, we develop an iterative structure-calculation protocol that exploits the power of integrative structural biology to characterize conformational ensembles.

**\*For correspondence:** teresa.carlomagno@oci.uni-hannover.de

**Competing interests:** The authors declare that no competing interests exist.

## Introduction

In a wide variety of cellular processes, ranging from biosynthesis to signalling and regulation of gene expression, RNA is chemically modified both co- and post-transcriptionally. All classes of RNA are modified, and RNA processing and editing mechanisms are highly conserved, with more than 140 chemical modifications supporting RNA function in all three domains of life (*Machnicka et al., 2013*). In rRNA, the most abundant modification is 2'-O-methylation, which impacts pre-rRNA processing, ribosome assembly and function. Functionally, 2'-O-methylation has been shown to protect RNA from ribonucleolytic cleavage (*Herschlag et al., 1993*), stabilize single base-pairs, act as a chaperone (*Helm, 2006*; *Williams et al., 2001*) and influence folding at high temperatures (*Kawai et al., 1992*). Nonetheless, the exact role of position-specific 2'-O-ribose methylation is mostly unknown.

Recent evidence shows that, while methylation sites are largely conserved and cluster in functionally important regions of the ribosome (*Decatur and Fournier, 2002*), the abundance of modified nucleotides is not uniform across species, or even across tissues. In humans, one third of methylated sites show variable levels of modification according to the cell-type (*Krogh et al., 2016*). The heterogeneous ribosome population resulting from these different methylation levels is consistent with the notion of specialized ribosomes that translate particular genes with improved efficiency (*Xue and Barna, 2012*). In agreement with its putative role in regulating translation, the complexity of rRNA 2'-O-methylation has increased with evolution: in bacteria, a protein enzyme catalyses 2'-O-methylation at a handful of rRNA sites, while in yeast and humans a small nucleolar ribonucleoprotein complex (the Box C/D snoRNP) uses a set of guide RNAs to deposit methyl groups in a sequence-specific manner at ~50 and 100 rRNA sites, respectively.

Besides their role in guiding 2'-O-methylation, Box C/D RNPs are involved in a variety of other functions, ranging from rRNA processing (for example, the U3 snoRNP, *Kass et al., 1990*) to RNA base acetylation (*Sharma et al., 2017*). Furthermore, nearly half of all human snoRNPs have no predictable rRNA targets, suggesting that they may have other roles within the cell (*Falaleeva et al., 2017*). Some of these so-called orphan snoRNPs have been associated with cancer and other diseases (*Gong et al., 2017*; *Williams and Farzaneh, 2012*).

The varying levels of methylation measured at different sites and the involvement of the Box C/D RNPs in processes other than methylation raise the question as to how the enzymatic activity is regulated or even silenced in the various Box C/D RNPs.

The lack of an in vitro reconstitution protocol yielding an active snoRNP currently precludes mechanistic and structural studies of the eukaryotic Box C/D complex. All structural and in vitro functional work to date has focused on the archaeal Box C/D sRNP (*Figure 1a*). The validity of this system as a proxy for the eukaryotic enzyme is established by their architectural similarity and comparable complexity of the rRNA methylation patterns (~115 rRNA methylation sites are predicted in *Pyrococcus furiosus*).

In archaea, Box C/D sRNPs consist of three proteins assembled around the guide sRNA (*Figure 1—figure supplement 1*). Within the guide RNA, the highly conserved box C/D sequence motif folds into the kink-turn (K-turn) (*Kiss-László et al., 1998*) structure and recruits the protein L7Ae (Snu13 and 15.5K in yeast and human, respectively) (*Moore et al., 2004*). By analogy, the less conserved box C'/D' motif has been proposed to fold into the kink-loop (K-loop) structure (*Nolivos et al., 2005*), which also binds L7Ae (*Gagnon et al., 2010*). The guide RNA–L7Ae complex binds the two C-terminal domains (CTDs) of the homodimer Nop5 (heterodimer Nop58–Nop56 in yeast and humans), which then recruits two copies of the methylation enzyme fibrillarin (Nop1 and fibrillarin in yeast and human, respectively) through its N-terminal domains (NTDs). The guide sRNA recognizes the rRNA substrate sequences at spacer regions located between boxes C and D' and between boxes C' and D; once bound to the substrate, it directs methylation to the fifth nucleotide upstream of either box D (substrate D) or D' (substrate D') (*Reichow et al., 2007*).

In the absence of substrate RNA (apo form), the archaeal Box C/D sRNP has been found to assemble mainly as a dimeric RNP, comprising four copies of each protein and two copies of the guide sRNA (*Bleichert et al., 2009*) (di-RNP, *Figure 1*). Upon saturation of the substrate RNA binding sites (holo form), two oligomeric states have been reported (*Figure 1—figure supplement 2*): the monomeric RNP (mono-RNP, *Lin et al., 2011*), containing two copies of each protein, one guide sRNA and two substrate RNAs (*Figure 1—figure supplement 2a*), and the dimeric RNP (di-RNP, *Lapinaite et al., 2013*), containing four copies of each protein, two guide sRNAs and four substrate RNAs (*Figure 1—figure supplement 2b*). Whether the existence of both mono- and di-RNP forms is merely a consequence of the different experimental set-ups in vitro or has a functional relevance in vivo remains an open question (*Yu et al., 2018*). In any case, the monomeric sRNP is believed to be a better representation of the eukaryotic system, as snoRNPs have never been shown to assemble into dimers, and the structure of the U3 snoRNP bound to a pre-ribosomal complex displays a mono-RNP architecture (*Cheng et al., 2017*).

The levels of methylation catalysed by sRNP complexes in vitro vary according to the substrate sequence. In early studies the efficiency of 2'-O-methylation in vitro was proposed to depend on the stability of the substrate–guide duplex and on the formation of an ideal A-form helical geometry close to the modification site (*Appel and Maxwell, 2007*). Using the *Pyrococcus furiosus (Pf)* sR26 guide RNA, whose corresponding sRNP methylates substrate D' more efficiently than substrate D, we demonstrated that methylation levels depend on — among other factors — the nature of the first base-paired nucleotide of the substrate (*Graziadei et al., 2016*). The observation that substrate D', with a 5'-uridine, displays good turnover in all conditions, while turnover of substrate D, with a 5'-guanosine, requires binding of substrate D' (*Graziadei et al., 2016*), led us to suggest that the nature of the last base-pair before the box D (or box D') regulates product dissociation. In agreement with the hypothesis that methylation levels are not exclusively dependent on the stability of the substrate–guide duplex, a recent study, which quantified site-specific rRNA methylation in two different human cell lines (*Krogh et al., 2016*), revealed that methylation levels in vivo do not correlate with either the number of base-pairs or the stability of the substrate–guide helix.

Here we demonstrate that the sequence of the substrate–guide duplex influences the affinity of fibrillarin for the substrate and that the extent of fibrillarin binding correlates with the efficiency of

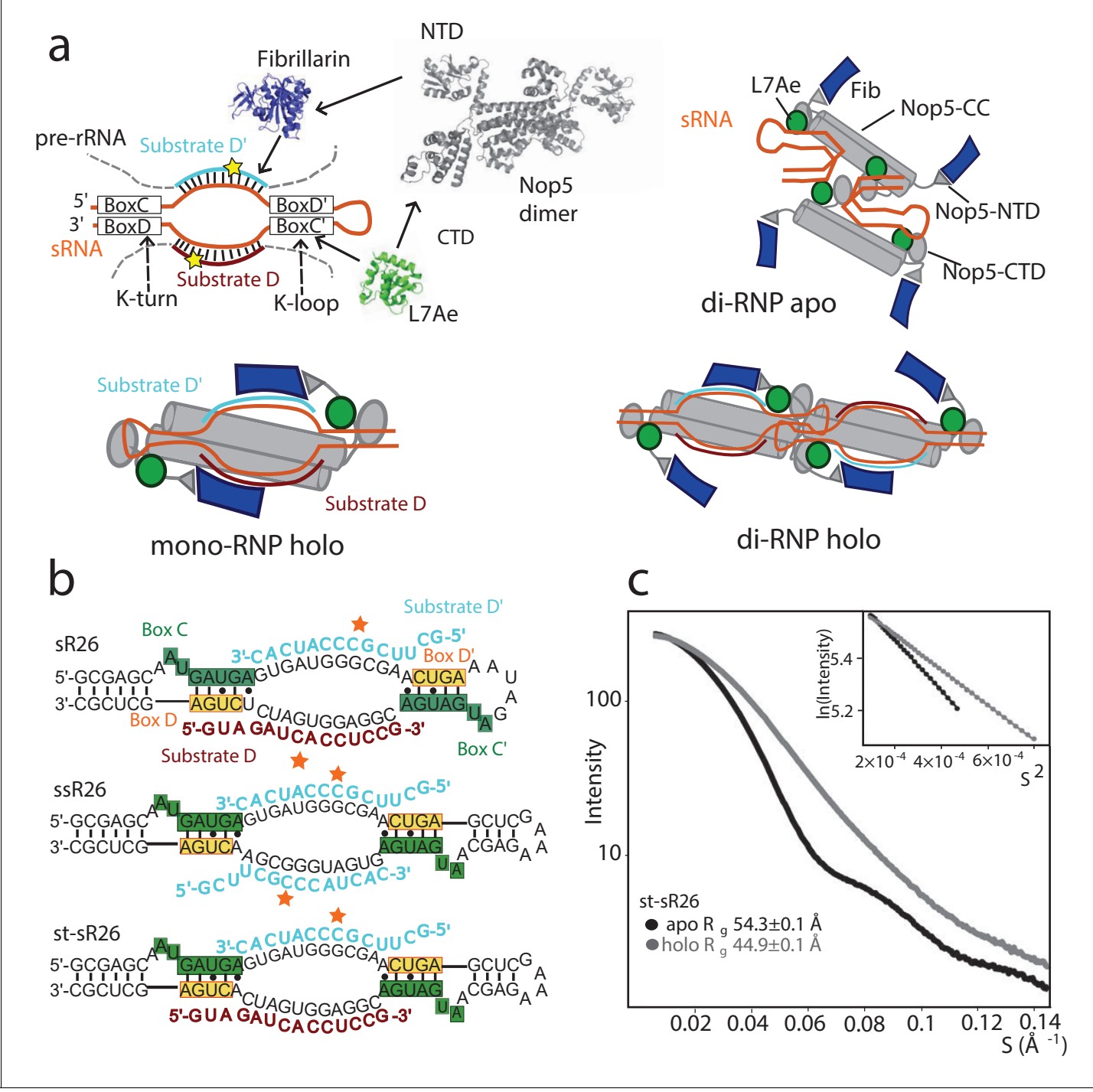

**Figure 1.** Oligomeric assembly states of the archaeal Box C/D RNP. (a) Top-left: molecular components of the archaeal Box C/D sRNP. Top-right: schematic model of the apo sRNP. Bottom-left: schematic model of the holo mono-RNP from *Lin et al. (2011)* Bottom-right: schematic model of the holo di-RNP from *Lapinaite et al. (2013)*. NTD: N-terminal domain; CTD: C-terminal domain; CC: coiled-coil. (b) Two RNA sequences (st-sR26 and ssR26) were derived from the *Pf* sR26 RNA and used to assemble the Box C/D sRNPs either in this (st-sR26) or previous studies (ssR26, *Lapinaite et al., 2013*). The sequence of st-sR26 is derived from the native sR26 RNA by substitution of the apical K-loop element with the more stable K-turn element. (c) SAXS curves with Guinier plots in the inserts of the Box C/D sRNPs reconstituted with st-sR26 before (apo) and after (holo) addition of 1.25 equivalents of each of substrate D and D' at a concentration of 2 mg/ml. The transition from an apo di-RNP to a holo mono-RNP is evident from the respective $R_g$ values (*Figure 1—figure supplement 4*). The data was collected at 40°C. All curves are scaled to the same forward scattering intensity. The online version of this article includes the following figure supplement(s) for figure 1:

**Figure supplement 1.** Conservation of the Box C/D RNP between archaea and yeast.

*Figure 1 continued on next page*

*Figure 1 continued*

**Figure supplement 2.** The mono- and di-RNP states of the archaeal holo sRNP.

**Figure supplement 3.** The sRNP assembled with sR26 has the same oligomerization behaviour as the sRNP assembled with st-sR26.

**Figure supplement 4.** Ranges of radii of gyration for the mono- and di-RNP states of the archaeal sRNP.

**Figure supplement 5.** Dependence of the oligomeric state of the holo sRNPs on the substrate-recognition sequence of the guide RNA.

**Figure supplement 6.** The SANS curves of $^2$H-Fib indicate the presence of either four or two copies of fibrillarin in the apo and substrate-loaded st-sR26 RNPs, respectively.

methylation. Using nuclear magnetic resonance (NMR), small angle X-ray (SAXS) and neutron (SANS) scattering data, we demonstrate that, in the context of the sRNP complex, the affinity of fibrillarin for the substrate depends on the RNA sequence beyond the methylation site. This difference in affinity is explained by the energetics of a global conformational transition of the sRNP from an inactive to an active state and provides a further route, besides the modulation of product dissociation described previously (*Graziadei et al., 2016*), to tune RNA methylation levels. To derive these results we developed an ensemble structure-calculation method that exploits the ability of integrative structural biology in solution to reveal and characterize conformational equilibria.

## Results

### Structure determination of the half-loaded mono-RNPs

To understand the reasons for the higher efficiency of substrate D' methylation as compared to substrate D in the *Pf* sR26 RNP we set out to determine the structure of the corresponding half-loaded sRNPs, bound to either substrate D or substrate D'. We used a stabilized version of the *Pf* sR26 guide RNA, where the apical K-loop has been substituted by a K-turn sequence (stabilized sR26, st-sR26, *Figure 1b*). This modification was necessary to ensure that the complex remains stably assembled over several days at 55˚C, as required by the NMR experiments, and does not affect the oligomerization state of the complex (*Figure 1* and *Figure 1—figure supplement 3*).

First, we determined the oligomerization state of the RNP complexes assembled with st-sR26 from their radius-of-gyration ($R_g$), measured by SAXS or SANS. To estimate the compatibility of experimentally determined $R_g$ values with the mono- or di-RNP assembly states, we evaluated the theoretical $R_g$ distributions of 5000 di-RNP models with randomized positions of the fibrillarin copies not bound to the RNA in both apo and holo (fully-loaded) conformations from *Lapinaite et al. (2013)*; *Figure 1—figure supplement 4*). We obtained a mean $R_g$ value of 55.9 Å with a standard deviation (SD) of 2.0 Å for the apo di-RNP and a mean $R_g$ of 58.1 ± 3.6 Å for the holo di-RNP. The SAXS curves of the apo sRNP assembled with st-sR26 (*Figure 1c*) correspond to a radius-of-gyration ($R_g$) of 54.3 Å, which is consistent with a di-RNP architecture (*Figure 1—figure supplement 4*). Addition of 1.25 molar equivalents of either substrate D or D' reduces the $R_g$ from 54.3 Å to 50.0 or 47.3 Å, respectively, with a further reduction to 45.0 Å, upon addition of both substrates (holo state) (*Graziadei et al., 2016*). These radii are no longer compatible with a di-RNP, demonstrating that both the half-loaded and holo st-sR26 complexes are mono-RNPs (*Figure 1—figure supplement 4*). The same transition from a di-RNP to a mono-RNP occurred for the Box C/D RNP assembled with sR26 upon substrate RNA binding (*Figure 1—figure supplement 3*). This is different from the holo complex assembled previously in our laboratory with the ssR26 RNA (symmetric and stabilized sR26), which contains two substrate D' RNA binding sites of the same sequence (*Figure 1b* and *Figure 1—figure supplement 2b*). The RNP assembled with ssR26 remained a di-RNP after saturation of the substrate RNA binding sites (*Lapinaite et al., 2013*).

Before embarking upon the structural study of the sRNPs containing st-sR26, we wanted to understand which elements are responsible for the different oligomerization states of the holo ssR26- and holo st-sR26-RNPs. The ssR26 and the st-sR26 RNAs differ only in the sequence of the guide RNA at the box D position, which in the case of ssR26 is identical to that of guide D'. Thus, we generated two additional guide RNAs with distinct D and D' sequences, st-sR26-1 and st-sR26-2: in st-sR26-1 (st-sR26-2), guide sequence D is a chimeric sequence, formed by the 5' half of st-sR26 guide D (st-sR26 guide D') and the 3' half of st-sR26 guide D' (st-sR26 guide D) (*Figure 1—figure supplement 5a*). Interestingly, the Box C/D enzyme containing st-sR26-1 maintained the di-RNP

architecture upon binding of either substrate RNAs, while the sRNP containing st-sR26-2 transitioned to the mono-RNP state (*Figure 1—figure supplement 5b*). Mutation of the last nucleotide of st-sR26-1 guide D to either C or U (A61C and A61U with complementary substrate D) did not perturb the di-RNP architecture (*Figure 1—figure supplement 5c*). We conclude that the guide sequence strongly influences the oligomerization state of the holo complex.

Further evidence of the monomeric state of half-loaded and holo st-sR26 complexes emerges from the P(r) distribution calculated from the SANS curve of the complexes assembled with $^2$H-fibrillarin in 42%:58% $D_2O:H_2O$ solvent: the number and relative intensities of the maxima are compatible with the presence of two fibrillarin copies but incompatible with the presence of four (*Figure 1—figure supplement 6*). As monomeric complexes, the substrate-loaded st-sR26 RNPs can serve as proxies for the eukaryotic snoRNP. As we showed previously (*Graziadei et al., 2016*), the sRNP assembled with this RNA catalyses the methylation of the substrate D' more efficiently than substrate D, in a similar manner to the native *Pf* sR26 RNP.

To investigate whether the difference in methylation efficiency of substrate D and D' correlates with structural differences, we assembled the Box C/D RNP with the st-sR26 guide RNA and saturated either its D or D' guide site (*Figure 1b*) to obtain two half-loaded mono-RNPs. We then determined their structures in solution, where the conformational dynamics of the complexes are preserved. The mono-RNPs are ~190 kDa in size and thus not amenable to standard structure determination by NMR. In this molecular-weight range, solution NMR focuses on methyl-group resonances, which have favourable relaxation properties and show strong signal intensity (*Sprangers and Kay, 2007*; *Tugarinov et al., 2003*) Thus, to solve the structure of the two half-loaded sRNPs, we used a combination of methyl-group NMR spectroscopy and small-angle scattering (see Methods and *Carlomagno, 2014*).

As in our earlier work on the fully-loaded di-RNP complex (*Lapinaite et al., 2013*), we started from the assumption that the interaction interface of the Nop5-CTD with the L7Ae–K-turn-RNA complex and that of the Nop5-NTD with fibrillarin do not change with respect to those observed in the respective crystal structures (*Liu et al., 2007*; *Xue et al., 2010*; *Aittaleb et al., 2003*). To validate this assumption we acquired two-dimensional $^1$H-$^{13}$C correlation spectra of fibrillarin and L7Ae labelled specifically at the methyl groups of Ile, Val and Leu residues (*Tugarinov and Kay, 2003*). The chemical shift perturbations measured for L7Ae in the Box C/D mono-RNP with respect to L7Ae in the L7Ae–K-turn-sRNA complex map to the previously described interface between L7Ae and the Nop5-CTD (*Xue et al., 2010*; *Figure 2—figure supplement 1*). Similarly, the chemical-shift perturbations measured for fibrillarin in the Nop5-NTD–fibrillarin complex with respect to free fibrillarin map to the interaction interface observed in previous crystal structures (*Aittaleb et al., 2003*). These CSPs are conserved in the Nop5–fibrillarin complex and in the apo Box C/D mono-RNP (*Figure 2—figure supplement 2*), demonstrating that fibrillarin interacts exclusively with the Nop5-NTD in all complexes.

We then used the signals from the L7Ae and fibrillarin methyl groups to measure paramagnetic relaxation enhancements (PREs). In this technique, a paramagnetic tag (spin-label) carrying an unpaired electron is coupled to a unique cysteine engineered on one protein subunit within the complex. The PREs elicited on the methyl groups of a second protein subunit by the unpaired electron are translated into distance restraints (*Battiste and Wagner, 2000*), which define the position and relative orientation of the two subunits in the complex. For the D-loaded (D'-loaded) mono-RNP, we collected a total of 407 (442) PREs using spin-labels on L7Ae-Q45C, L7Ae-E58C/C68S, L7Ae-C68, Nop5-E196C, Nop5-D247C and Nop5-S343C while observing the methyl resonances of fibrillarin and on Nop5-E65C while observing the methyl resonances of L7Ae (*Figure 2—figure supplement 3a*). The PRE data were validated by means of intra-molecular PREs within the rigid fibrillarin module (*Figure 2—figure supplement 4*). The excellent fit between the experimental PRE intensity ratios and those predicted from the known distances confirms the reliability of the PRE-derived inter-molecular distances.

A second class of structural restraints was derived from SANS curves acquired with contrast-matching. In these experiments one or more proteins in the complex are $^2$H-labelled and contribute to the observed scattering signal, while the scattered intensity of the unlabelled proteins is masked by the solvent, which is prepared as a 42%:58% $D_2O:H_2O$ mixture. A combination of such datasets provides sufficient information to restrain the relative position of several molecules within a multi-subunit complex. In our case we acquired SANS curves for $^2$H-L7Ae, $^2$H-Nop5, $^2$H-Fib, $^2$H-RNA, $^2$H-

Fib/$^2$H-RNA and $^2$H(70%)-Nop5/$^2$H-RNA in 42%:58% $D_2O$:$H_2O$ (*Figure 2—figure supplement 3b*). In addition, we also collected SAXS curves, which report on the shape of the entire complexes.

These data were then incorporated into a structure-calculation protocol adapted from that developed in our previous study (*Lapinaite et al., 2013*) (for a description of the adapted protocol, refer to Methods and *Figure 3—figure supplement 1*). We used the conformations of the modules L7Ae–K-turn-sRNA–Nop5-CTD and Nop5-NTD–fibrillarin observed in previous crystal structures, and restricted our conformational search to the relative orientations of the three domains of Nop5, the conformation of the sRNA in parts other than the K-turn motifs and A-form helices and the relative positions of the two copies of each protein in the mono-RNP.

## Conformation of the half-loaded mono-RNPs in solution

The methyl-group NMR spectrum of fibrillarin in the apo RNP assembled with st-sR26 is identical to the spectrum of the RNP assembled with ssR26 (*Figure 2a*, left panel). This was expected, as in both di-RNPs all four fibrillarin copies are far from the RNA and thus their chemical shifts are independent of the RNA sequence used to assemble the complex.

Methyl groups are rather sparse in the protein surfaces involved in recognition of the RNA backbone; in the RNA-bound form of fibrillarin, only the methyl groups of V35, I82, V110, L114, I117, V151 and V185 are expected to be within 8 Å of the RNA, while only V110 should be closer than 5 Å. Therefore, the chemical shift perturbations (CSPs) for fibrillarin upon RNA binding should be few and relatively small in magnitude. As expected, the methyl-group NMR spectrum of the substrate-bound RNPs showed only moderate CSPs; nonetheless, these were mainly localized in the spectral region containing V110, V151 and V185, thus confirming that fibrillarin recognizes the substrate D'–guide duplex (*Figure 2a*, right panel).

Further evidence of substrate–guide recognition by fibrillarin was provided by the PRE data. As shown in *Figure 2b* for substrate D', upon fibrillarin binding to the substrate–guide duplex (on-state, upper left), the Nop5-E65C spin-label (red) comes close to one L7Ae copy and would lead to PRE intensity-ratios of less than 0.8 for the L7Ae-ILV residues shown as yellow spheres. In contrast, when fibrillarin is not bound to the substrate–guide duplex (off-state, upper right), the Nop5-E65C spin-label is far from L7Ae and cannot cause any PRE attenuation of L7Ae peaks. Thus, the low PRE intensity-ratios observed experimentally for the methyl groups of the residues marked in yellow (*Figure 2b*, bottom) indicates the presence of conformers in which fibrillarin is bound to the substrate–guide duplex.

In an half-loaded mono-RNP, one fibrillarin copy is necessarily in the off-state, due to the lack of the corresponding substrate; the second fibrillarin copy could be either stably bound to the substrate–guide duplex (yielding a complex in the [on,off]-conformation) or exchanging between the on- and off-states (corresponding to the RNP exchanging between the RNP [on,off]- and [off,off]-conformations, *Figure 2c*). The NMR data are qualitatively compatible with both scenarios, as the broad line-widths and the overlap of the fibrillarin NMR peaks that show the largest CSPs upon RNA binding preclude a quantitative analysis of the magnitude of the CSPs in terms of relative proportions of the two conformations. Thus, we decided to consider both scenarios in the interpretation of the structural data.

## Structure calculations

To determine the [on,off]- and [off,off]-conformations of both the substrate D- and D'-loaded sRNPs, we adapted our previously developed structure-calculation protocol (*Lapinaite et al., 2013*). We initially performed two structure calculations per complex: in the first calculation, we imposed the restraint that one fibrillarin copy is in contact with the corresponding substrate–guide duplex, while the other copy is not ([on,off]-state); in the second calculation, we left both fibrillarin copies free to adopt any position compatible with the PRE data ([off,off]-state). We then recursively binned the PRE-derived distance-restraints into two sets, according to their compatibility with the the [on,off]- or [off,off]-conformations (*Figure 3—figure supplement 1*). The majority of restraints were found to be consistent with both states and therefore appeared in both sets. One notable exception is the set of PRE restraints derived from the methyl-groups of L7Ae in the presence of spin-labelled Nop5-E65C, which are compatible only with fibrillarin being in contact with the substrate–guide duplex (*Figure 2b*). In total, we performed four structure-calculation runs, two for each of the half-loaded

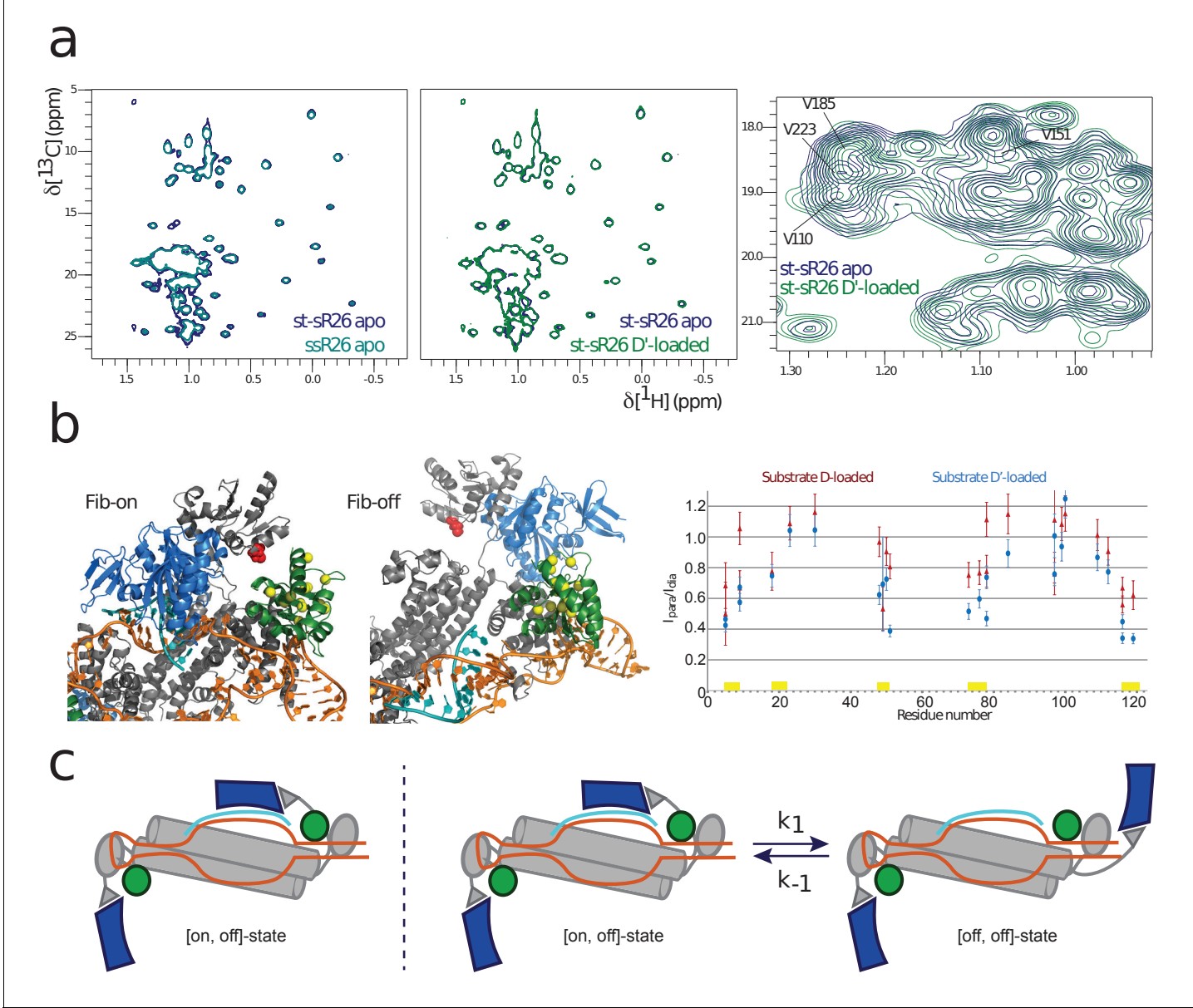

**Figure 2.** NMR and SAS of the half-loaded st-sR26 RNPs. (a) Left, overlay of ILV-methyl $^1$H-$^{13}$C spectra of fibrillarin in the apo ssR26 (turquoise) and apo st-sR26 (blue) RNPs. In both di-RNPs, all four fibrillarin copies are distant from the RNA and the two spectra are identical. Middle, overlay of ILV-methyl $^1$H-$^{13}$C spectra of fibrillarin in the apo st-sR26 (blue) and substrate D'-loaded st-sR26 (green) RNPs. Right, expanded view of the overlay of ILV-methyl $^1$H-$^{13}$C spectra of fibrillarin in the apo st-sR26 (blue) and substrate D'-loaded st-sR26 (green) RNPs. (b) Left, structural snapshots of the on- (left) and off- (right) states of one fibrillarin copy in the substrate D'-loaded mono-RNP. Upon binding of fibrillarin to the substrate–guide duplex, the Nop5-E65C spin-label (red) comes close to one L7Ae copy (green), leading to PRE intensity-ratios below 0.8 for the L7Ae-ILV residues shown as yellow spheres. In contrast, when fibrillarin is in the off-state (right), the Nop5-E65C spin-label is far from L7Ae and cannot induce any PRE-mediated attenuation of peak intensities. Colour-code as in *Figure 1*. Right, PRE effects ($I_{para}/I_{dia}$, ratio of the peak intensities when the spin-label is in the paramagnetic and diamagnetic state, respectively) of the Nop5-E65C tag on the L7Ae-ILV peaks in the substrate D-bound (red) and substrate D'-bound (blue) mono-RNPs. The yellow bars indicate the residues represented as yellow spheres in the left panel. (c) Left, cartoon representation of the [on,off]-conformer of the substrate D'-loaded mono-RNP; right, cartoon representation of the conformational equilibrium between the [on,off]- and [off,off]-conformers of the same complex.

The online version of this article includes the following figure supplement(s) for figure 2:

**Figure supplement 1.** L7Ae in the Box C/D mono-RNP maintains the previously determined interaction interfaces with Nop5-CTD.

**Figure supplement 2.** Fibrillarin in the Box C/D mono-RNP maintains the previously determined interaction interfaces with Nop5-NTD.

**Figure supplement 3.** Schematic summary of the experimental data.

**Figure supplement 4.** Validation of the PRE-derived distances.

complexes. The [on,off]-conformations were compatible with nearly all PRE-derived restraints (400 out of 407 for the substrate D-loaded and 436 out of 442 for the substrate D'-loaded RNP, respectively), while the [off,off]-conformations were compatible with 364 and 414 restraints for the substrate D- and substrate D'-loaded RNP, respectively.

Each individual structure calculation proceeded through a global and a local search stage. At each stage, the total and distance-restraint energies as well as the back-calculated fits to the SANS curves were used for structure selection. The two final structure ensembles corresponding to the [on,off]-states (*Figure 3*) are defined to a precision of better than 2.5 Å (root-mean-square-deviation, RMSD, of the protein Cα and RNA P atoms, excluding flexible regions). When compared to the existing structure of the holo mono-RNP from *Sulfolobus solfataricus* (PDB entry 3pla, *Lin et al., 2011*), the substrate D- and substrate D'-loaded complexes show a reasonable similarity (*Figure 3— figure supplement 2*). All major features of the substrate-bound site are conserved: the RNA-guide sequences lie on the coiled-coil Nop5 domain at an angle of about 70° and the C-terminal tip of L7Ae is in proximity to the short Nop5 β-sheet 77–79 and α-helix 64–73. However, the solution structures differ from the crystallographic structure in many details, demonstrating that the sRNP architecture is flexible enough to adapt to different guide- and substrate-RNAs. As expected, a significant divergence from the structure of PDB entry 3pla is observed in the substrate-unbound half of the complexes.

Importantly, neither the [on,off]- nor the [on,on]-ensemble are able to reproduce the combination of PRE and SAS data satisfactorily for each of the substrate D- or the substrate D'-loaded RNPs. The PRE intensity-ratios measured for the Nop5-NTD-E65C mutant on the methyl-groups of L7Ae indicate the presence of conformers in the [on,off]-state. In agreement with this, the [on,off]-structures of *Figure 3* reproduce the PRE data reasonably well both for the substrate D- and substrate D'-loaded complexes (*Figure 3—figure supplements 3* and *4*). However, these structures are unable to fit the $^2$H-Fib SANS, $^2$H-Fib/$^2$H-RNA SANS and SAXS curves in a satisfactory manner (*Figure 3— figure supplement 5*). Thus, the combination of PRE and SAS data is incompatible with a single state for each of the substrate D- or substrate D'-loaded RNPs, but rather reveals the presence of conformational ensembles.

## Conformational ensembles

Because the SAS data that are in disagreement with the [on,off]-conformations of *Figure 3* all report on the position of the fibrillarin copies in the complexes, we deduced that the conformational equilibria present in solution must be related to the position of fibrillarin. Different types of conformational equilibria are conceivable. In the simplest scenario, only the fibrillarin in the off-state samples multiple conformations, with the second fibrillarin remaining stably in the on-state; in a more complex scenario, the second fibrillarin copy may sample both the on- and off-states (in addition to the conformational flexibility of the fibrillarin copy in the off-state).

To represent both scenarios and obtain structural ensembles compatible with both PRE and SAS experimental data, we developed an ensemble scoring protocol (*Figure 3—figure supplement 1b*, Methods). For both the substrate D- and substrate D'-loaded RNPs, we used representative structures of the [on,off]- and [off,off]-state ensembles (*Figure 3*) — defined as the structure closest to the mean structure — as starting points to generate four sets of ~4000 conformations, in which the positions of the Nop5-NTD–fibrillarin units not bound to the substrate–guide duplex were randomized, in order to account for their flexibility. We then used a pseudo-genetic algorithm to select ensembles of either exclusively [on,off]-conformers or of both [on,off]- and [off,off]-conformers that best fit the PRE data, as well as the $^2$H-Fib and $^2$H-Nop5 SANS, $^2$H-Fib/$^2$H-RNA SANS, $^2$H(70%)-Nop5/$^2$H-RNA SANS and SAXS curves (*Figure 3—figure supplement 1*).

## Conformational ensemble of the substrate D'-loaded sRNP

Despite the reasonable fit of the PRE intensity ratios of the substrate D'-loaded sRNP with the representative structure of the [on,off]-conformers of (*Figure 3*; *Figure 3—figure supplement 3*), the larger $R_g$ of the experimental $^2$H-Fib SANS curve with respect to the theoretical one indicated the presence of conformers where the two copies of fibrillarin are more distant from each other than in this set of [on,off]-conformers (*Figure 3—figure supplement 5*).

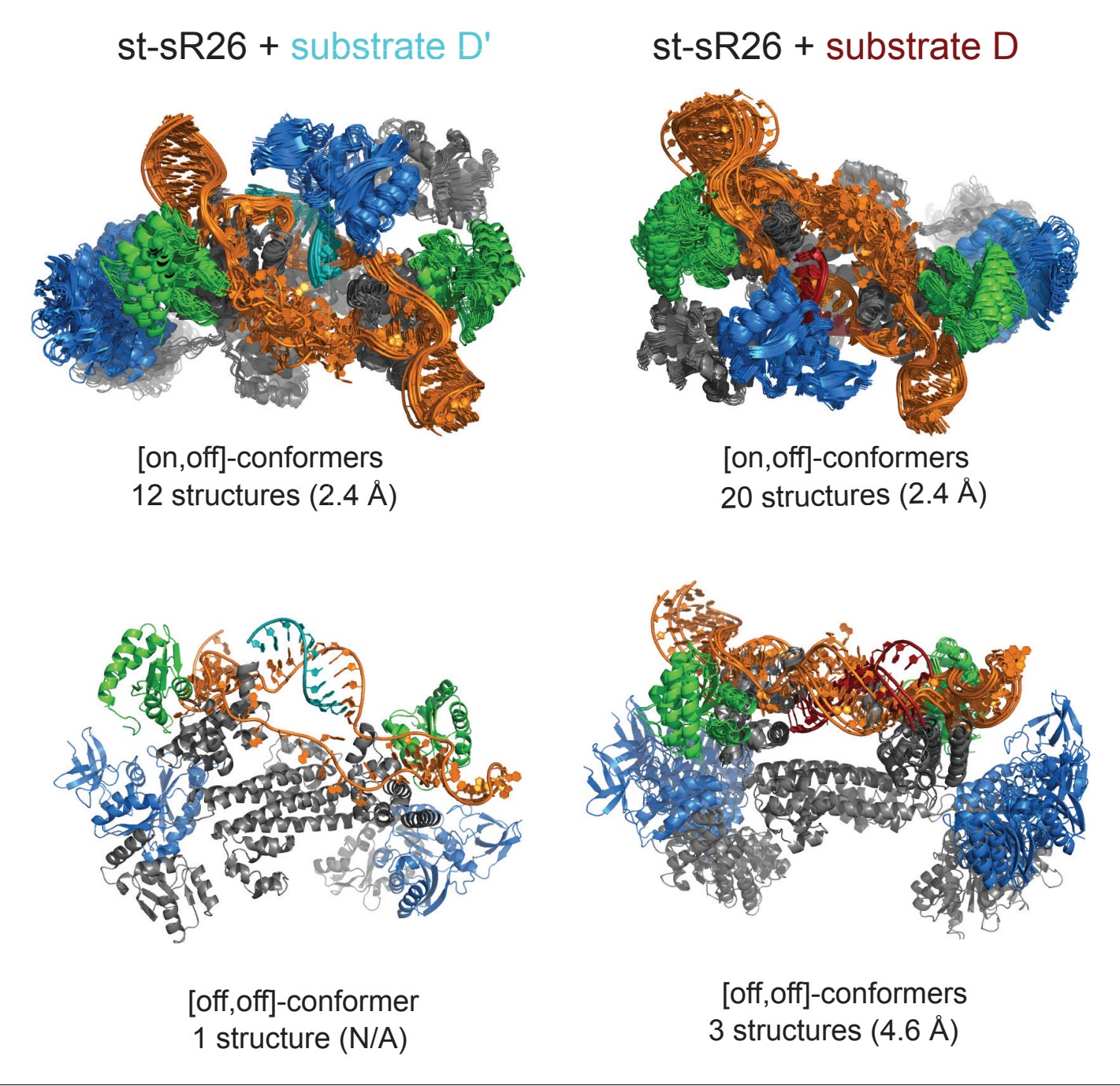

**Figure 3.** Ensembles of structures in agreement with the experimental data for the [on,off]- and [off,off]-states of substrate D'- and substrate D-loaded sRNPs. The RMSD values of each ensemble (in parentheses) are calculated as the average of the RMSD values of the ensemble structures with respect to the structure closest to the mean over the Cα and P atoms of the protein and RNA structured domains, including the fibrillarin units not bound to the RNA. Colour-code as in *Figure 1*.

The online version of this article includes the following figure supplement(s) for figure 3:

**Figure supplement 1.** Structure-calculation algorithms.

**Figure supplement 2.** Structures of the half-loaded sRNPs in the [on,off]-state.

**Figure supplement 3.** Fit of individual [on,off]- or [off,off]-conformers to the PRE data of the substrate D'-loaded sRNP.

**Figure supplement 4.** Fit of individual [on,off]- or [off,off]-conformers to the PRE data of the substrate D-loaded sRNP.

**Figure supplement 5.** Fit of individual [on,off]- or [off,off]-states to the SAS data.

We thus set out to improve the fit to the experimental data by deriving mixed ensembles containing both [on,off]- and [off,off]-conformers using the ensemble scoring protocol described above. The resulting best-fit ensembles contained 66 ± 8% [on,off]-conformers and showed a much improved fit to both the SAXS and $^2$H-Fib SANS curves (*Figure 4*). The agreement between experimental and predicted PREs also improved (*Figure 5*).

To verify that an acceptable fit to the experimental data requires the combination of both [on,off]- and [off,off]-conformers in the structural ensemble, we repeated the ensemble scoring protocol

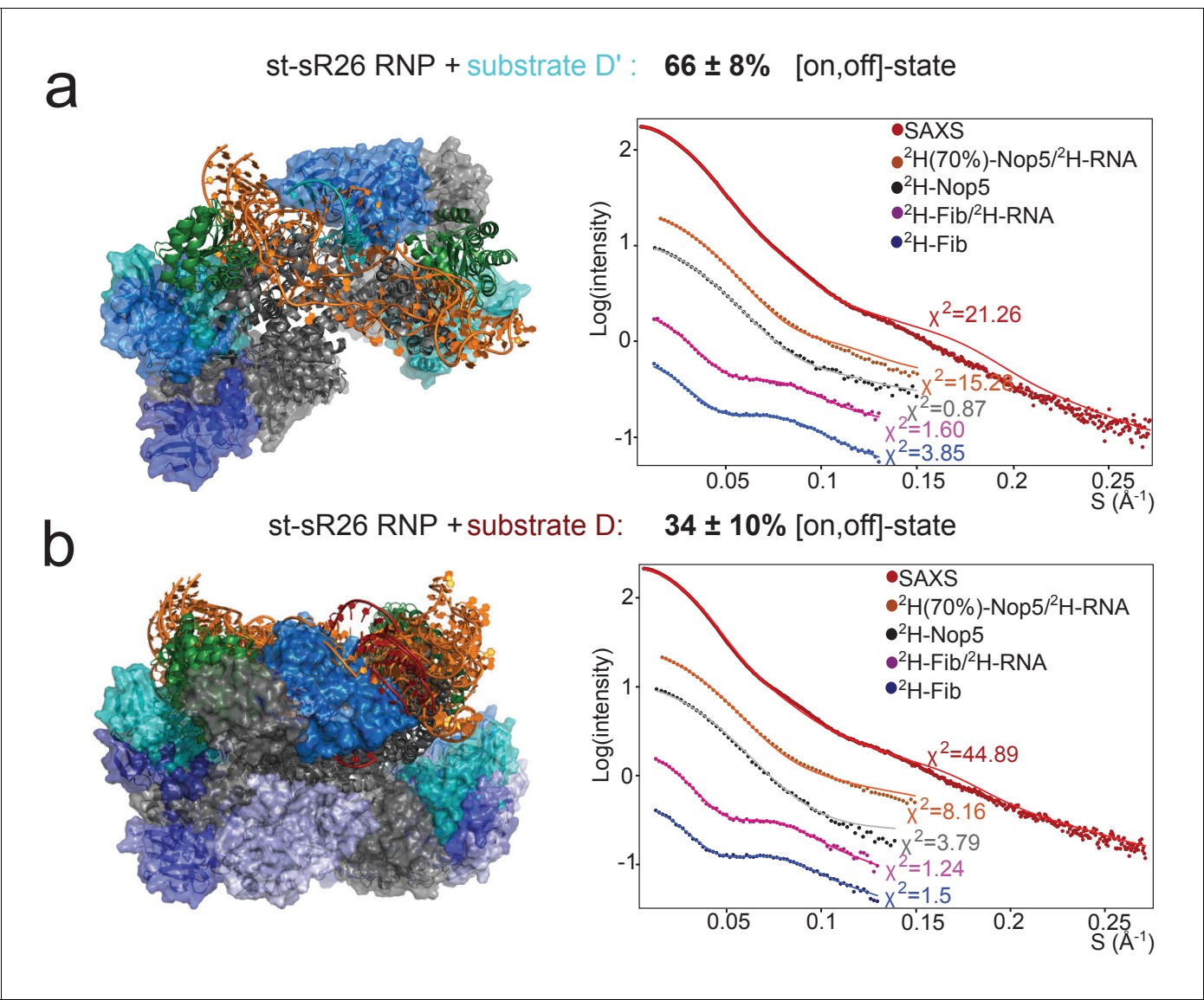

**Figure 4.** Fibrillarin binds the substrate–guide duplex more strongly in the substrate D'-loaded sRNP. (**a**) The structural ensemble selected by the pseudo-genetic scoring algorithm (Methods) for the substrate D'-loaded sRNP, containing two [on,off]-state and one [off,off]-state conformers, with fibrillarin shown in shades of blue. The fits to the experimental SAS curves are shown on the right. All SANS curves were measured in 42%:58% $D_2O$: $H_2O$. (**b**) Structural ensemble selected by the pseudo-genetic scoring algorithm for the substrate D-loaded sRNP, containing three [on,off]-state and eight [off,off]-state conformers. In both **a** and **b**, the mean and standard deviation of the percentage of [on,off]-state structures in the three top-scoring ensembles across three independent scoring runs is shown in the title. The structural ensembles yield much better agreement with the SAS curves than do the individual [on,off]- and [off,off]-state structures (*Figure 3—figure supplement 5*).

The online version of this article includes the following figure supplement(s) for figure 4:

**Figure supplement 1.** Fit to the SAS data of ensembles of only [on,off]- or [off,off]-conformers after randomisation of the position of the fibrillarin copy in the off-state.

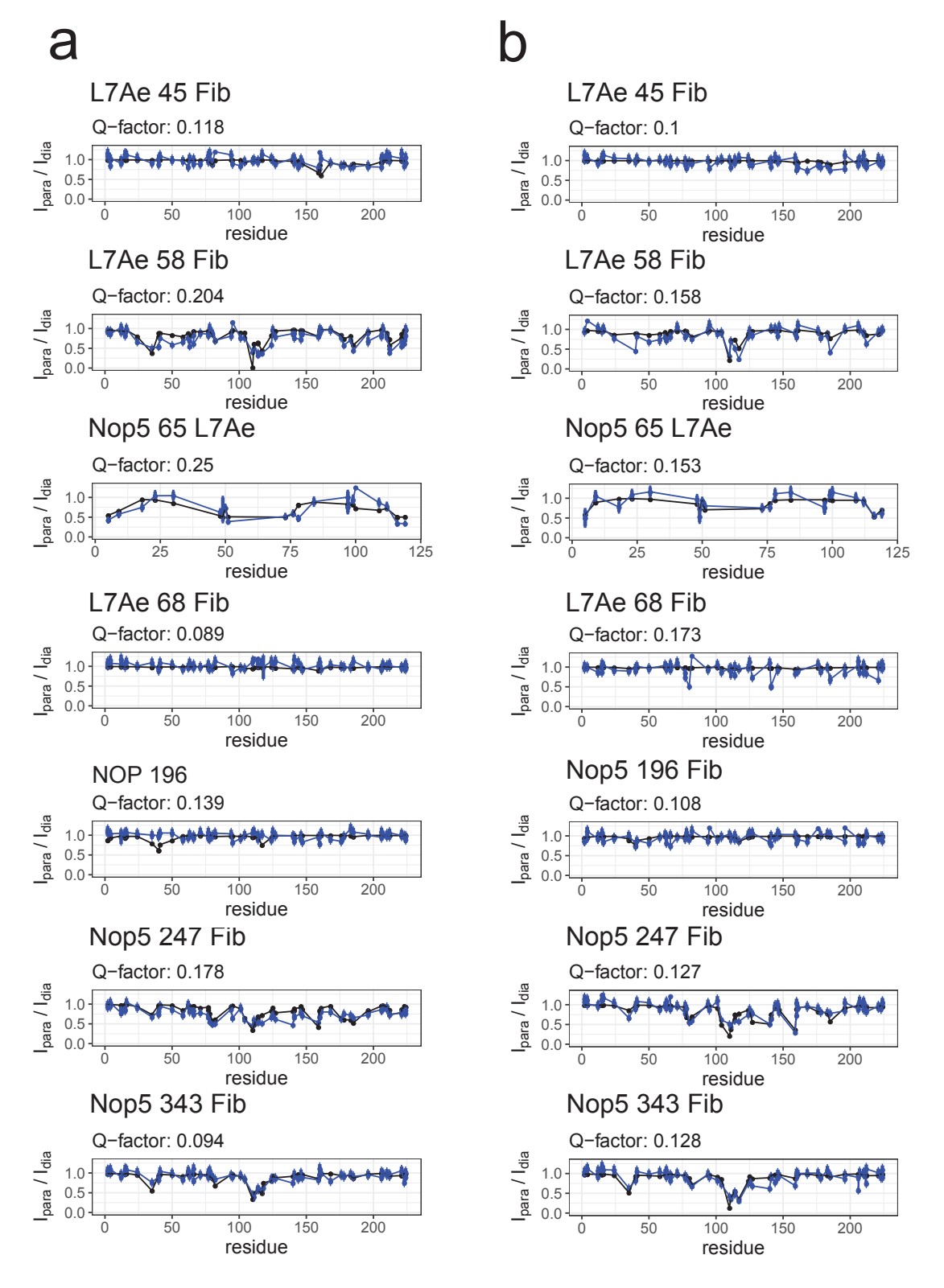

**Figure 5.** Fit of the ensemble structures representing the substrate–loaded RNPs to the PRE data. (a) Comparison of $I_{para}/I_{dia}$ ratios back-calculated from the selected ensemble of conformers of the substrate D'-loaded st-sR26 RNP shown in *Figure 4a* (blue) with the experimental ratios (black). The reported Q-factors were calculated as recommended by *Clore and Iwahara (2009)*. In the title of each panel the first name indicates the spin-labelled protein, the number indicates the position of the spin-label and the second name indicates the protein whose ILV methyl groups were detected. (b)

*Figure 5 continued*

Comparison of $I_{para}/I_{dia}$ ratios back-calculated from the selected ensemble of conformers of the substrate D-loaded st-sR26 RNP shown in *Figure 4b* (blue) with the experimental ratios (black). The structural ensembles yield better or similar agreement with the PRE data than do the individual [on,off]- and [off,off]-state structures *Figure 3—figure supplements 3* and *4*).

The online version of this article includes the following figure supplement(s) for figure 5:

**Figure supplement 1.** Fit to the PRE data of ensembles of only [on,off]- or [off,off]-conformers after randomisation of the position of the fibrillarin copy in the off-state.

selecting from only [on,off]- or [off,off]-conformers. The fit to the SAS curves remained unsatisfactory for both these ensembles (*Figure 4—figure supplement 1*), with the [on,off]-ensemble yielding a poor fit to the ²H-Fib SANS curve and the [off,off]-ensemble being unable to reproduce the SAXS curve. In addition, the fit of the [on,off]-ensemble to the PRE data (*Figure 5—figure supplement 1a*) remained inferior to that of the ensemble containing both [on,off]- and [off,off]-structures.

## Conformational ensemble of the substrate D-loaded sRNP

The higher values of the PRE intensity-ratios measured for the L7Ae methyl-groups in the presence of the spin-labelled Nop5-NTD-E65C mutant in the substrate D-loaded mono-RNP as compared to the substrate D'-loaded mono-RNP indicated that the proportion of fibrillarin bound to the substrate–guide duplex is lower for the mono-RNP loaded with substrate D than for that loaded with substrate D'. Accordingly, the combination of PRE and SAS data could not be fit with an ensemble consisting of [on,off]-conformers only, as the $\chi^2$ value of the SAXS curve remained as poor as that obtained with a single [on,off]-conformer (>250) (*Figure 4—figure supplement 1*). Conversely, ensembles containing only [off,off]-conformers failed to reproduce the PRE dataset of the complex containing spin-labelled Nop5-E65C (*Figure 5—figure supplement 1b*).

When we fitted the PRE and SAS data with ensembles consisting of both [on,off]- and [off,off]-conformers, we could reproduce all experimental data satisfactorily with a population of [on,off]-conformers of 34 ± 10% (*Figures 4* and *5*).

## Fibrillarin binds preferentially to substrate D'

The combination of the NMR and SAS data demonstrated the existence of a conformational equilibrium between [on,off]- and [off,off]-conformers for both substrate D- and substrate D'-loaded RNPs. The ensemble of conformations representing the substrate D'-loaded sRNP (*Figure 4*) contained a reproducibly higher proportion of conformers with fibrillarin in the [on,off]-state (66 ± 8%) than did the ensemble representing the substrate D-loaded sRNP (34 ± 10%), as was expected from the stronger PRE effects induced on L7Ae by the Nop5-E65C paramagnetic tag for the substrate D'-loaded sRNP (*Figure 2b*). Thus, despite the lack of sequence-specific interactions with the RNA, fibrillarin binds more strongly to the substrate D'–guide duplex than to the substrate D–guide duplex in the context of the Box C/D RNP.

This observation prompted us to analyse in more detail the structural differences between the [on,off]-states of the substrate D- and D'-loaded RNPs, as well as their stability in a 150-ns molecular-dynamics (MD) simulation. In the [off,off]-state, both half-loaded RNPs display a regular A-form helix of 11 base-pairs formed by the guide and substrate RNAs and positioned far from the Nop5 coiled-coil domains. The geometry of this helix was given as a restraint in the structure calculations, because of the perfect complementarity of the substrate–guide sequences over these 11 nucleotides. Binding of fibrillarin pushes the substrate–guide duplex towards the Nop5 coiled-coil domain, thereby perturbing the base-pairing at the substrate 3' end (*Figure 6*). This observation is in agreement with a recent study, reporting that a substrate–guide duplex of only 10 base-pairs results in the highest level of in vitro methylation for a *S. solfataricus* enzyme (*Yang et al., 2016*). During the 150-ns MD trajectory of the D'-loaded complex, the two base-pairs at the 3' end of substrate D' are disrupted and the Nop5 α10 helix and its flanking loops form many electrostatic contacts with the RNA (*Figure 7*). In addition, W319 forms a face-to-face interaction with the no-longer base-paired G15 of the guide RNA. In contrast, in the substrate D-loaded complex, only one base-pair is melted at the 3' end of the substrate (the second last), fewer new contacts are formed between the protein and the RNA and some other contacts are lost during the simulation (*Figures 6* and *7*). Furthermore,

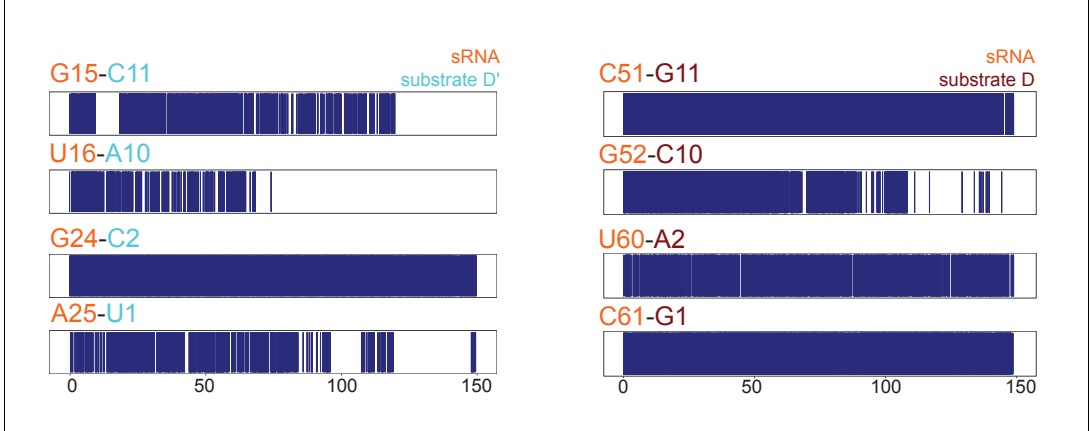

**Figure 6.** Substrate–guide duplex hydrogen-bonds throughout the molecular dynamics runs. Plots showing the hydrogen bonding pattern across substrate–guide duplex 3' and 5' ends in the substrate D'-bound (left) and substrate D-bound (right) sRNPs over two 150-ns molecular dynamics simulations. A blue line indicates the presence of at least two hydrogen bonds between the corresponding bases. The numbering is according to *Figure 1*.

in the substrate D'-loaded complex the A–U base-pair at the 5' end of substrate D' iwas often disrupted during the simulation, allowing for the formation of electrostatic contacts between E289 and A25/C5 and K290 and U4 (*Figure 7*). Conversely, in the substrate D-loaded complex, the C–G base-pair at the 5' end of the substrate remains stable throughout the simulation (*Figure 6*). The first-base paired nucleotide of substrate D is kept in place by hydrogen bonds between G22 of the unpaired guide and its sugar backbone. In agreement with our MD simulations, *Yang et al. (2016)* demonstrated that high levels of methylation occur for a substrate–guide duplex length of 8–10 base pairs.

We conclude that the stability of the fibrillarin-bound form depends on a delicate balance between the loss of entropy due to fibrillarin localization, and the positive and negative enthalpy changes associated with base-pair melting and formation of new protein–RNA contacts, respectively. Given that:

$$\left( G_{on,off}^{D'} - G_{off,off}^{D'} \right) + \left( G_{on,off}^{D} - G_{off,off}^{D} \right) = RT \left( ln\frac{p_{on,off}^{D'}}{p_{off,off}^{D'}} - ln\frac{p_{on,off}^{D}}{p_{off,off}^{D}} \right) \tag{1}$$

where $G_{on,off}^{D'}$ $\left( G_{on,off}^{D} \right)$ and $G_{off,off}^{D'}$ $\left( G_{off,off}^{D} \right)$ are the free energies of the substrate D' (D)-loaded complex in the [on,off]- and [off,off]-states, respectively, and $\frac{p_{on,off}^{D'}}{p_{off,off}^{D'}}$ $\left( \frac{p_{on,off}^{D}}{p_{off,off}^{D}} \right)$ is the ratio of the populations of the substrate D' (D)-loaded complex in the [on,off]- and [off,off]-states, we can calculate that the difference between the ΔG values for the [off,off]→[on,off] transition of the substrate D'- and substrate D-loaded complexes in the st-sR26 RNP is only 0.86 ± 0.55 kcal/mol. This small value suggests that fine differences in the stability of the substrate–guide helices may regulate the affinity of fibrillarin for the methylation site and thus the fractional population of active enzyme.

## Discussion

2'-O-rRNA methylation is one of the most extensive modification processes occurring during ribosome synthesis and maturation. The strong conservation of the methylation sites over different species, together with the lethal effect of methylation suppression, led to the conclusion that methylation is a constitutive modification of functional ribosomes. However, rRNA methylation has recently been proposed to exert a regulatory function by generating an heterogeneous ribosome population with differential methylation levels (*Erales et al., 2017*).

2'-O-methylation is implemented by the Box C/D RNP enzyme through an RNA-guided catalysis. In addition to methylation, Box C/D complexes are involved in a plethora of other functions related

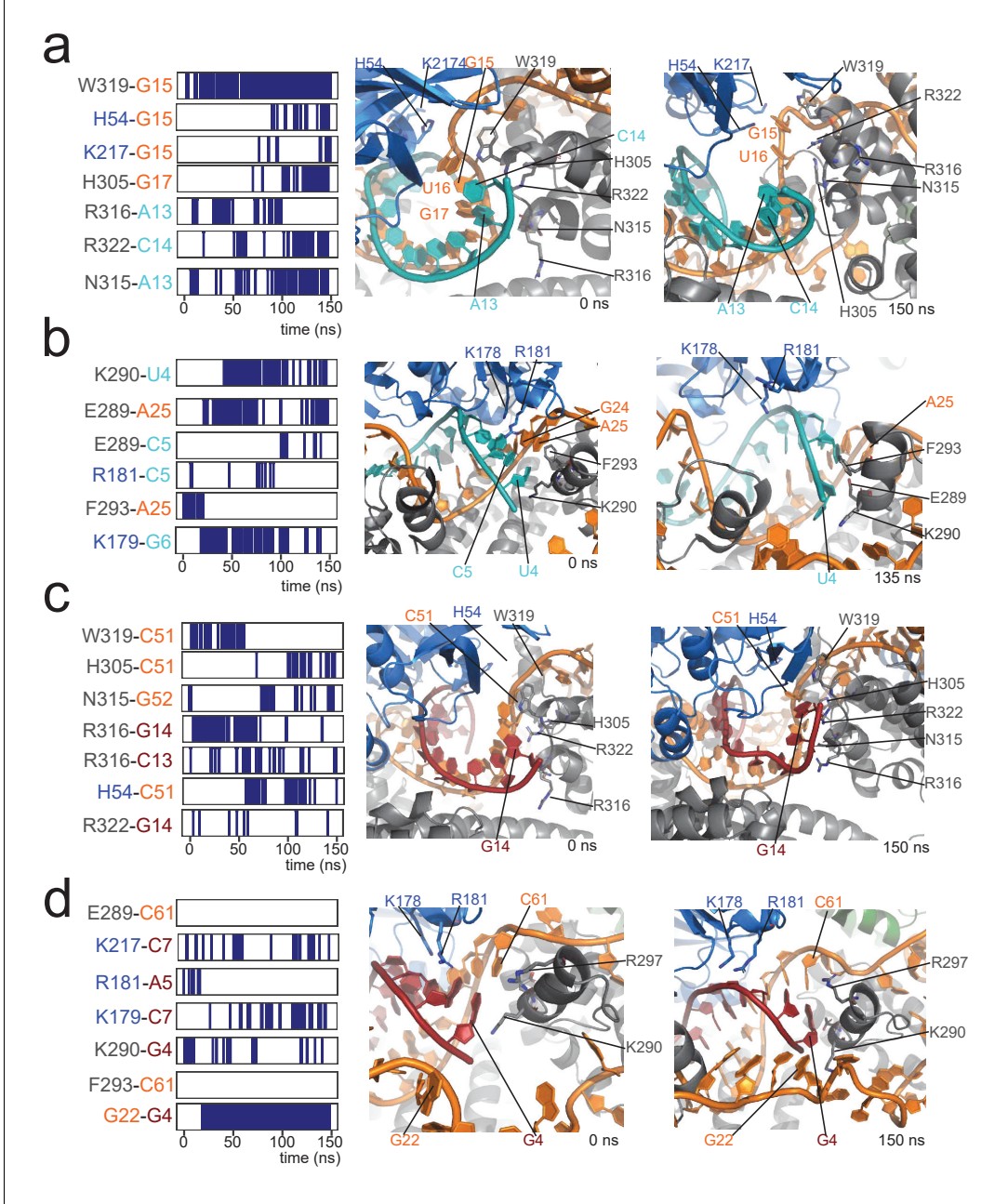

**Figure 7.** Contacts between proteins and the 5' and 3' ends of the substrate–guide duplex in a 150-ns molecular dynamics run. (a) Protein–RNA contacts at the 3' end of the substrate–guide duplex in the [on,off]-state of the substrate D'-bound RNP. Each line marks the presence of a contact between the two residues under consideration. Black and blue indicate amino acids of Nop5 and fibrillarin, respectively; orange and cyan indicate nucleotides of the sRNA and substrate D', respectively. The numbering of the RNA is as in *Figure 1*. Contacts are H-bonds between polar amino-acid side-chains and polar atoms of the nucleotide (as defined in CPPTRAJ within Amber); hydrophobic interactions involving aromatic amino acid side chains and base rings (with a distance cut-off of 4.0 Å between the centres of the rings); electrostatic contacts between polar amino acid side chains and the RNA phosphorus atoms (with a distance cut-off of 4.0 Å between the polar group and the P atom). The interacting amino acids and nucleotides are displayed in the structural panel in the middle (starting structure) and on the right (structure towards the end of the simulation). (b) Protein–RNA contacts at the 5' end of the substrate–guide duplex in the [on,off]-state of the substrate D'-bound RNP. (c) Protein–RNA contacts at the 3' end of the substrate–guide duplex in the [on,off]-state of the substrate D-bound RNP. Only the second last base-pair melts, leading to a lower number of protein–RNA contacts as compared to the substrate D'-bound RNP. (d) Protein–RNA contacts at the 5' end of the substrate–guide duplex in the [on,off]-state substrate D-bound RNP. Both the RNA secondary structure and the position of the RNA relative to the proteins remain constant throughout the simulation, without formation of new protein–RNA contacts.

to RNA processing. In the context of the multiple roles of Box C/D complexes, the question arises as to how Box C/D RNPs distinguish whether the RNA substrate bound to the guide sequence should be methylated and to what extent.

To address this question, we studied the structure-function relationship of Box C/D RNPs in solution through a combination of NMR and SAS data. Using the archaeal Box C/D sRNP, we determined the solution-state structures of the half-loaded substrate D- and substrate D'-bound mono-RNPs. Instead of a single, well-defined conformational state, we found that the copy of fibrillarin at the substrate-loaded site exchanges between the substrate-bound and unbound states, with the substrate D'-loaded complex displaying a higher population of the methylation-competent [on,off]-state than the substrate D-loaded RNP. Accordingly, the substrate D'-loaded RNP achieves higher levels of methylation (*Graziadei et al., 2016*). The existence of dynamic equilibria between substrate-bound and unbound conformers of fibrillarin has not been detected by X-ray crystallography, which instead selects for the most ordered conformation.

Our results suggest that the proportion of the methylation-competent complex is subtly tuned by the free-energy difference between the active [on,off]- and inactive [off,off]- conformations (and possibly also by the kinetics of transition, on which our structural data at equilibrium do not provide any information). Recognition of the RNA ribose by fibrillarin is accompanied by a loss in entropy at the junction between the Nop5-NTD and the Nop5 coiled-coil and between the box C/D (or box C'/D') RNA elements and the substrate–guide duplex. In addition, upon fibrillarin binding, the substrate–guide helical structure must deviate from the ideal A-form geometry, in order to adapt to the proteins. This is particularly evident at the 3' end of the substrate, where any base-pair beyond the tenth is melted (*Yang et al., 2016*; *Figure 6*). These energetically costly events are compensated by the formation of contacts between fibrillarin and the RNA backbone, as well as by contacts between the Nop5-CTDs and fibrillarin and the two ends of the substrate–guide duplex. MD simulations showed that the substrate D'–guide duplex is less stable at the substrate 3' end than the substrate D–guide duplex, and that the melting of the last two base-pairs results in a large number of protein–RNA contacts. These appear to stabilize the [on,off]-state in the substrate D'-loaded RNP, suggesting that the exact sequence of the 3' end segment of the substrate RNA influences the fractional population of the active conformation of the RNP. However, as we are unable to detect the RNA signals of the 190 kDa RNP by NMR spectroscopy in solution, we cannot exclude the possibility that in vivo other RNA elements, involving for example the overhang at the substrate 5' and 3' ends, could also play a role in stabilizing the [on,off]-state, as previously suggested (*Appel and Maxwell, 2007*).

In conclusion, methylation efficiency appears to be regulated by a complex interaction network depending on the substrate rRNA sequence beyond the methylation site. We propose that, together with substrate turnover (*Graziadei et al., 2016*), the ability of different substrate–guide duplexes to shift the position of the equilibrium between the [off,off]- and [on,off]-state conformers modulates the level of methylation at distinct rRNA sites. When the difference in the free energies of the active and inactive enzyme states is small, the correspondingly variable ratio between the populations of the active and inactive conformations provides a mechanism to tune the activity level. When the free-energy difference is large and positive, the population of the methylation-competent conformation becomes vanishingly small and the Box C/D RNP loses its capacity to catalyse methylation. This situation could be the basis for supporting functions of the Box C/D complexes that are unrelated to methylation and thus may not require fibrillarin to bind the RNA (for example, the U3 snoRNP, which guides the formation of the central pseudoknot in 18S rRNA).

To calculate the structural models of the substrate D- and substrate D'-loaded RNPs we developed a novel hybrid structure-calculation protocol that fits a combination of NMR and SAS data to an ensemble of conformations. The application of integrative structural biology approaches is particularly relevant to the detection of inter-domain dynamics of RNP complexes, as the different types of structural data are sensitive to conformational changes in different ways. In this case, the combination of NMR PRE data and SAS data was essential for revealing the equilibrium between RNA-bound and RNA-unbound fibrillarin states. The computational workflow developed here allows interpretation of hybrid structural data in terms of structural ensembles, rather than as individual conformations. The protocol proceeds in a step-wise fashion, where the structural ensemble becomes progressively well-defined while increasing the demand on the quality of the fit between predicted and experimental data. We anticipate the methodology developed here to be generally applicable to modular enzymes undergoing domain reorientation during catalysis.

# Materials and methods

## Key resources table

| Reagent type (species) or resource | Designation | Source or reference | Identifiers | Additional information |
|---|---|---|---|---|
| Strain, strain background (*Escherichia coli*) | BL21 (DE3) | EMBL protein expression facility | NA | |
| Strain, strain background (*Escherichia coli*) | BL21 Rosetta 2 | Merck Millipore | Cat #71400–3 | |
| Recombinant DNA reagent | pETM-11 Fibrillarin (plasmid) | *Lapinaite et al. (2013)* | | Nterminal His6 + TEV site |
| Recombinant DNA reagent | pETM-11 Nop5 (plasmid) | *Lapinaite et al. (2013)* | | Nterminal His6 + TEV site; L113K V223E mutant. Codon-optimised synthetic gene (GeneArt) |
| Recombinant DNA reagent | pETM-11 Nop5 E65C (plasmid) | *Lapinaite et al. (2013)* | | Mutation of pETM-11 Nop5 |
| Recombinant DNA reagent | pETM-11 Nop5 E196C (plasmid) | *Lapinaite et al. (2013)* | | Mutation of pETM-11 Nop5 |
| Recombinant DNA reagent | pETM-11 Nop5 D247C (plasmid) | *Lapinaite et al. (2013)* | | Mutation of pETM-11 Nop5 |
| Recombinant DNA reagent | pETM-11 Nop5 S343C (plasmid) | *Lapinaite et al. (2013)* | | Mutation of pETM-11 Nop5 |
| Recombinant DNA reagent | pETM-11 L7Ae (plasmid) | *Lapinaite et al. (2013)* | | Nterminal His6 + TEV site |
| Recombinant DNA reagent | pETM-11 L7Ae Q45C (plasmid) | *Lapinaite et al. (2013)* | | Mutation of pETM-11 L7Ae also carrying C68S mutation |
| Recombinant DNA reagent | pETM-11 L7Ae E58C (plasmid) | *Lapinaite et al. (2013)* | | Mutation of pETM-11 L7Ae also carrying C68S mutation |
| Sequence-based reagent | st-sR26 | *Graziadei et al. (2016)* | In vitro transcribed RNA | |
| Sequence-based reagent | st-sR26-1 | This paper | In vitro transcribed RNA | Method section: RNA synthesis |
| Sequence-based reagent | st-sR26-1 substrate | This paper | In vitro transcribed RNA | Method section: RNA synthesis |
| Sequence-based reagent | st-sR26-1 A61C | This paper | In vitro transcribed RNA | Method section: RNA synthesis |
| Sequence-based reagent | st-sR26-1 A61U | This paper | In vitro transcribed RNA | Method section: RNA synthesis |
| Sequence-based reagent | st-sR26-2 | This paper | In vitro transcribed RNA | Method section: RNA synthesis |
| Sequence-based reagent | st-sR26-2 substrate | This paper | In vitro transcribed RNA | Method section: RNA synthesis |
| Sequence-based reagent | sR26 | *Graziadei et al. (2016)* | In vitro transcribed RNA | |
| Sequence-based reagent | ssR26 | *Lapinaite et al. (2013)* | In vitro transcribed RNA | |
| Commercial assay or kit | TLAM-ILVproS labelling | NMR-Bio | NA | |

*Continued on next page*

*Continued*

| Reagent type (species) or resource | Designation | Source or reference | Identifiers | Additional information |
|---|---|---|---|---|
| Chemical compound, drug | Iodoacetoamido-PROXYL | Sigma-Aldrich | Cat # 253421–25 MG | |
| Chemical compound, drug | (methyl-13C, 99%; 3,3-D2, 98%) α-ketobutyric acid | Cambridge Isotope Labs | CDLM-7318-PK | |
| Chemical compound, drug | (3-methyl-13C, 99%; 3,4,4,4-D4, 98%) α-ketoisovaleric acid | Cambridge Isotope Labs | CDLM-7317-PK | |
| Chemical compound, drug | [3–2 H2,4–2H, 5–13C, 5'–2 H3]-a-ketoiso-caproate | *Lichtenecker et al. (2013)* | | |
| Software, algorithm | CNS | This paper | | Method section: Structure calculation and selection. Adaptation of protocol from *Lapinaite et al. (2013)* |
| Software, algorithm | Python-based SAS-PRE scoring algorithm | This paper | | |
| Software, algorithm | ATSAS 2.7.5 | *Petoukhov et al., 2012* | | |
| Software, algorithm | Python-based SAS-PRE scoring algorithm | This paper | | Method section: Ensemble Scoring |

## Protein expression, labelling and purification

L7Ae (UniProtKB accession code Q8U160), Nop5 (Q8U4M1) and archaeal fibrillarin (Q8U4M2) were expressed, purified and reconstituted with sRNAs as described previously (*Graziadei et al., 2016*). Nop5 was expressed with the L113K and V223E mutations in order to prevent the formation of aggregates. Deuterated proteins were expressed in 100% $D_2O$ M9 minimal medium using $^2$H-glycerol as the sole carbon source. Deuterated proteins with $^1$H,$^{13}$C-labelled ILV methyl groups were produced following protocols developed in the Kay laboratory (*Tugarinov and Kay, 2003*). Stereospecific pro-S $^1$H,$^{13}$C-labelling of valine and leucine methyl groups was obtained by expression with the appropriate metabolic precursor according to the specifications of the manufacturer (TLAM-I$^{δ1}$LV$^{proS}$; NmrBio). Leucine-specific labelling was achieved using the protocol described by *Lichtenecker et al. (2013)*. All NMR samples were assembled with $^2$H-Nop5, and, in the case of $^1$H,$^{13}$C -ILV methyl-labelled L7Ae, with both $^2$H-Nop5 and $^2$H-fibrillarin. The $^2$H(70%)-Nop5 sample for SANS experiments was obtained by expression in 100% $D_2O$ M9 minimal medium with $^1$H-glucose as the sole carbon source; deuteration levels for this sample were verified by MALDI mass spectrometry.

## RNA synthesis

Guide-RNAs were produced by in vitro transcription from double-stranded plasmid DNA templates using T7 RNA polymerase produced in-house and rNTPs (Roth). RNAs were purified by denaturing 12–20% polyacrylamide gel electrophoresis, and extracted by electro-elution. For $^2$H-RNA samples, RNA synthesis was performed using $^2$H-labelled rNTPs (Silantes).

st-sR26: 5'-GCGAGCAAUGAUGAGUGAUGGGCGAACUGAGCUCGAAAGAGCAAUGAUGACG-GAGGUGAUCACUGAGCUCGC-3' st-sR26-1: 5'-CGAGCAAUGAUGAGUGAUGGGCGAACUGAGCUCGAAAGAGCAAUGAUGACGGAGGGGCGAACUGAGCUGCG-3'

st-sR26-2: 5'-CGAGCAAUGAUGAGUGAUGGGCGAACUGAGCUCGAAAGAGCAAUGAUGAGUGAUGUGAUCACUGAGCUGCG-3' sR26: 5'-GCGAGCAAUGAUGAGUGAUGGGCGAACUGAAAUAGUGAUGACGGAGGUGA UCUCUGAGCUCGC-3'

Substrate RNAs for st-sR26 were produced in-house using synthetic DNA oligonucleotides:
Substrate D': 5'-GCUUCGCCCAUCAC-3'

Substrate D: 5′-GUAGAUCACCUCCG-3′
st-sR26-1 substrate D: 5′-GUAUCGCCCCUCCG-3′
st-sR26-2 substrate D: 5′-GUAGAUCACAUCAC-3′

## Transfer of NMR methyl-group assignments

In the free state, fibrillarin methyl resonances were stereospecifically assigned by means of 3D NOE-SY–$^{13}$C-HMQC spectra, acquired on ILV and ILV$^{proS}$-labelled samples, in combination with 3D TOC-SY–$^{13}$C-HMQC spectra and by comparison to the NOEs expected from the fibrillarin structure. The assignment was transferred stepwise from the free fibrillarin to the Nop5-NTD–fibrillarin complex, the Nop5–fibrillarin complex and finally to the full Box C/D complex. For the ILV-labelled Nop5-NTD–fibrillarin complex, we also acquired a 3D NOESY–$^{13}$C-HMQC spectrum; for all complexes we acquired $^{13}$C-HMQC spectra on ILV-labelled, ILV$^{proS}$-labelled and L-labelled samples. For the ILV-labelled Nop5–fibrillarin complex, pairings of HMQC peaks from the diastereotopic methyl-groups of leucine and valine residues were verified with the assistance of a 3D experiment in which the $^1$H and $^{13}$C resonances of the methyl groups were correlated with the $^{13}$C resonances of the directly bonded methine carbon (Cγ and Cβ for leucine and valine residues, respectively), thereby allowing methyl-pairs to be identified from their common methine resonance. The pulse-sequence for this experiment comprises an out-and-back magnetization-transfer-pathway starting and ending on the methyl protons, using COSY-type transfers between the methyl and methine carbons and constant-time chemical-shift evolution periods for both indirect $^{13}$C dimensions.

## PRE measurements

Mutants were generated following the QUIKCHANGE-XL protocol (Agilent Technologies) and purified in the presence of 5 mM β-mercaptoethanol in order to prevent disulfide bond formation. For L7Ae, the native C68 was mutated to serine prior to the introduction of cysteine residues at other sites. The purified protein was then buffer exchanged into 50 mM NaPi, 500 mM NaCl, pH 6.6 using a HiPrep 26/10 desalting column (GE Healthcare) and eluted directly into tubes containing a 10-fold molar excess of the 3-(2-iodoacetamido)-PROXYL radical (Sigma-Aldrich) in the dark. The spin-labelling reaction was allowed to proceed overnight at room temperature. Spin-labelled proteins were used for complex reconstitution; the free spin-label was removed during the gel-filtration step. The final reconstitution step was carried out in 100% D$_2$O buffer (50 mM NaPi, 500 mM NaCl, pH 6.6), prior to concentration with a 10 kDa-cutoff Amicon centrifugal concentrator (Merck Millipore).

All substrate-loaded sRNPs were obtained by addition of 1.25 molar equivalents of substrate RNA. This ratio yields full saturation of the substrate RNA-binding sites of the guide RNA. We verified this by monitoring the appearance of peaks indicative of free RNA (sharp peaks) in one-dimensional $^1$H spectra of the sRNP upon addition of increasing concentrations of substrate RNA. Sharp peaks began to appear after a 1:1 molar ratio of substrate:guide RNA was reached.

$^{13}$C-HMQC spectra were acquired on Bruker Avance 800 and 850 MHz spectrometers, equipped with TCI cryoprobes, at 55°C with sample concentrations between 10 and 40 μM (2–8 mg/ml). Diamagnetic spectra were recorded after reduction of the spin-label by addition of ascorbic acid to a final concentration of 5 mM.

All spectra were processed using apodization with an exponential function in order to preserve Lorentzian line-shapes. Peaks were fitted with the program FUDA (http://www.ucl.ac.uk/hansen-lab/fuda/) assuming Lorentzian line-shapes. When necessary, overlapped peaks were fitted as groups. The fitted volumes and line-widths were then converted into peak-heights. The heights in the paramagnetic and diamagnetic states were used to calculate the distance between the nitroxide group of the paramagnetic tag and the respective methyl-group (see below).

The diamagnetic R$_2$ rates corresponding to the transverse relaxation rates of $^1$H single-quantum coherence (R$_2^{diaH}$) and $^1$H-$^{13}$C multiple-quantum coherence (R$_2^{diaHC}$) of each individual peak were quantified using the pulse-schemes from the Kay laboratory (*Tugarinov and Kay, 2006*; *Tugarinov and Kay, 2013*), modified to remove the fast-relaxing-component purging-element. Relaxation delays were 0, 2, 3, 4, 6, 7, 10 and 16 ms for fibrillarin, and 0, 2, 3, 4, 6, 7 and 10 ms for L7Ae. The peak-heights were fitted to a mono-exponential decay function to extract R$_2^{diaH}$ and R$_2^{diaHC}$.

In order to derive the correlation-time for the electron-nucleus interaction vector, $\tau_C$, we quantified paramagnetic ($I_{para}$: oxidized, paramagnetic state of the spin-label) and diamagnetic ($I_{dia}$: reduced, diamagnetic state of the spin-label) peak-heights corresponding to known distances within fibrillarin in complexes reconstituted with the Fib-R109C mutant. For L7Ae, we used known distances between the Nop5-CTD and L7Ae in complexes reconstituted with the Nop5-S343C mutant. The ratios of peak-heights were converted into PREs ($\Gamma_2$), using *Equation 2* and the $R_2^{diaHC}$ and $R_2^{diaH}$ rates measured for the respective peaks.

$$\frac{I_{para}}{I_{dia}} = \frac{exp(-\Gamma_2 t_{HMQC}) R_2^{diaH} R_2^{diaHC}}{\left(R_2^{diaH} + \Gamma_2\right)\left(R_2^{diaHC} + \Gamma_2\right)} \tag{2}$$

where $t_{HMQC}$ represents the magnetization transfer time in the HMQC sequence (7.6 ms). As this equation is non-invertible, $\Gamma_2$ was derived by plotting the simulated bleaching ratio, $I_{para}/I_{dia}$, as a function of $\Gamma_2$ for a given set of diamagnetic rates, with the experimental errors on $I_{para}/I_{dia}$, $R_2^{diaH}$ and $R_2^{diaHC}$ used to determine the upper and lower bounds of the derived PRE. These PREs were then used as restraints in the protocol developed in the Clore Lab (*Iwahara et al., 2004*), which optimizes an ensemble of multiple spin-label conformations in combination with $\tau_C$. For L7Ae, we used isoleucine resonances only. The minimization was run using the recommended 'obsig' setting for the weighting of the different PREs. After minimization of 20 structures, $\tau_C$ was 51.8 ± 5.7 ns for fibrillarin and 50.4 ± 9.4 ns for L7Ae.

For a given value of $\tau_C$, distances $r$ between the unpaired electron and the methyl protons were extracted from the equation:

$$r = \sqrt[6]{\frac{K}{\Gamma_2}\left(4\tau_C + \frac{3\tau_C}{1 + \omega^2\tau_c^2}\right)} \tag{3}$$

where K is a constant ($1.23 \times 10^{-23}$ cm$^6$ s$^{-2}$) and $\omega$ is the proton Larmor frequency in rad/s. The errors on the distances were again estimated by using the errors in $\tau_C$, experimental $I_{para}/I_{dia}$ ratios and $R_2$ rates to yield upper and lower bounds on a calibration curve. A lower-bound of 10% was used for the errors of $I_{para}/I_{dia}$, as recommended by Battiste & Wagner (Battiste & Wagner, 2000). A lower-bound of 2 Å was imposed for the errors on the distances in order to account for tag flexibility. Finally, a minimum error of −4 Å was used as lower bound for the distances extracted from the PRE ratios in the calculation of the [on,off]-structures, to account for the possibility that the methyl group of only one fibrillarin copy is close the paramagnetic tag: in this case, the effective distance of the methyl group of the one fibrillarin copy to the paramagnetic tag would be smaller than the distance calculated from the sum of the two overlapping fibrillarin peaks (one with PRE intensity-ratios < 0.8 and one with PRE intensity-ratios close to 1).

In the structure calculations (CNS), distances were imposed from the nitrogen atom of the nitroxide group of the paramagnetic tag to the carbon atoms of fibrillarin methyl groups. For L7Ae, where stereospecific assignment of LV methyl groups was not available, the distance restraint was imposed to both methyl group carbons with an 'OR' statement. For complexes with both fibrillarin copies positioned away from the RNA, the same set of distance restraints was imposed on each fibrillarin copy; for complexes with one fibrillarin copy close to the RNA, distance restraints were imposed with an 'OR' statement.

## Small-angle X-ray scattering (SAXS)

Box C/D sRNPs reconstituted in 50 mM NaPi pH 6.6, 500 mM NaCl were recorded at 40°C and concentrations varying from 0.4 to 5 mg/ml, unless otherwise specified. In most experiments a temperature of 40°C instead of 55°C was used for SAXS measurements due to the difficulty in collecting data with high salt concentrations at the higher temperature. For all measurements, 2 mM dithiothreitol (DTT) was added to mitigate radiation damage. Data collection was performed at the ESRF bioSAXS beamline BM29 with exposure of 10 frames each of 1 s duration. The curves were compared, merged, and the buffer contribution subtracted by the beamline software BsxCube (*Pernot et al., 2013*). Forward scattering intensity I(0) values were normalized relative to an ideal protein in an ideal solution, and were reported as 288, 194, 215 and 197 for the apo st-sR26 RNP, the substrate D'-bound st-sR26 RNP, the substrate D-bound st-sR26 RNP and the holo st-sR26 RNP, respectively, all

at 5 mg/ml. The $R_g$ and I(0) values were extracted according to the Guinier approximation using PRI-MUS in ATSAS 2.7.5 (*Konarev et al., 2003*). All $R_g$ values were computed using an s.$R_g$ upper limit of 1.3 (where s is the modulus of the scattering vector), as recommended for globular particles.

To estimate the compatibility of the experimentally determined $R_g$ values with the mono- or di-RNP assembly states, we evaluated the theoretical $R_g$ distributions of 5000 di-RNP models in both apo and holo conformations from *Lapinaite et al. (2013)* and 500 half-loaded mono-RNP models generated in both [on,off] and [off,off]-states using the torsion-angle simulated-annealing protocol described below. The apo di-RNP showed a mean $R_g$ value of 55.9 Å with a standard deviation (SD) of 2.0 Å; the holo di-RNP showed a mean $R_g$ of 58.1 ± 3.6 Å; the [on,off]-state of the mono-RNP showed a mean $R_g$ of 44.7 ± 1.4 Å; and the [off,off]-state of the mono-RNP showed a mean $R_g$ of 48.5 ± 1.7 Å (*Figure 1—figure supplement 3*).

## Small-angle neutron scattering (SANS)

$^2$H-L7Ae, $^2$H-Nop5, $^2$H-fibrillarin, $^2$H-RNA, $^2$H-fibrillarin/$^2$H-RNA and $^2$H(70%)-Nop5/$^2$H-RNA samples were measured in 50 mM NaPi pH 6.6, 500 mM NaCl, 42%:58% $D_2O$:$H_2O$ solutions, in order to mask the contribution of the $^1$H-proteins. The curves corresponding to $^2$H-L7Ae, $^2$H-Nop5, $^2$H-RNA and $^2$H(70%)-Nop5/$^2$H-RNA were acquired at D22 at the Institute Laue Langevin (ILL, Grenoble, France), with a neutron wavelength of 6 Å. The $^2$H-fibrillarin and $^2$H-fibrillarin/$^2$H-RNA curves were acquired at KWS-1 at JCNS (Munich, Germany) (*Feoktystov et al., 2015*) with a neutron wavelength of 5 Å. Both instruments were configured with sample-detector distances of 4 m and collimation lengths of 4 m. Data reduction and radial integration were done with standard procedures using beamline-specific software. Buffer subtraction was done in PRIMUS. Pair-wise distance-distribution functions P(r) were calculated from experimental data using GNOM in ATSAS 2.7.5 (*Svergun, 1992*). All SANS curves were acquired at 55°C.

## Structure calculation and selection

Structures were calculated using an adapted version of the protocol described in *Lapinaite et al. (2013)*; *Nilges, 1995* according to the workflow described in *Figure 3—figure supplement 1*. The starting st-sR26 RNA structures, bound to either substrate D or substrate D', were generated in separate calculation runs using restraints to impose an A-form helical geometry on the substrate–guide duplex, and to yield the appropriate K-turn structures. Starting protein conformations were generated from the PDB entry 3nmu and assembled into two L7Ae–Nop5–Fib protomers, in which the L7Ae–Nop5-CTD and Nop5-NTD–Fib interaction interfaces of 3nmu were preserved, but not the relative orientation of the Nop5-NTD and CTD, which were randomised. The two copies of the protomers within the sRNP were separated and randomly rotated with respect to each other. The building-blocks L7Ae–Nop5-CTD, Nop5-NTD–Fib and the Nop5 coiled-coil domain were kept rigid throughout the calculations. Structures were calculated for both the substrate D- and substrate D'-loaded sRNPs. For each sRNP the proteins and RNA were subjected to two sets of parallel torsion-angle simulated-annealing procedures; one included a set of restraints positioning one fibrillarin copy on the methylation site of the substrate–guide duplex ([on,off]-state); in another no restraints were imposed between fibrillarin and the RNA ([off,off]-state). The conformational sampling was driven by PRE-derived distance restraints, distance restraints positioning the two L7Ae–Nop5-CTD modules onto the RNA K-turns and a loose distance restraint between the centres of mass of the two L7Ae modules (90 ± 15 Å), which was derived from the P(r) curve of $^2$H-L7Ae in 42%:58% $D_2O$:$H_2O$. Restraints positioning the Nop5-α9' helix between the two guide regions (from Nop5-K301 and K304 to the phosphate backbone of the nucleotide linking the K-turn and substrate–guide helix) were also used. With this set up, we started an iterative procedure, to generate two lists of PRE-derived distance-restraints compatible with either the [on,off]- or [off,off]-state. 500 structures were calculated per iteration. At the end of each iteration, restraint violations were evaluated: restraints violated by more than 10 Å in either set of calculations were classified, eliminated from that particular set, but kept in the other. After 5 iterations, this led to two restraint-lists per sRNP, corresponding to the [on,off]- and [off,off]-states of the sRNP.

With these four sets of restraints (two for the substrate D-loaded and two for the substrate D'-loaded sRNP), four separate runs of torsion-angle simulated-annealing calculations were performed; we generated 2500 structures per run, using the settings described in *Lapinaite et al. (2013)*.

The fitness of the experimental SAS and PRE data with respect to the calculated structures was assessed by calculation of the $\chi^2$ statistic (*Equation 4*) and by visual inspection of fits between back-calculated and experimental data:

$$\chi^2 = \frac{1}{N}\sum_{i=1}^{N}\left[\frac{I_{exp}(s_i) - cI_{calc}(s_i)}{\sigma(s_i)}\right]^2 \tag{4}$$

where $I_{calc}$ represents the back-calculated data-point ($I_{para}/I_{dia}$ intesity-ratios or SAS intensities), $I_{exp}$ is the corresponding experimental value, N is the number of experimental points, $\sigma$ represents the experimental error and c is the scaling factor:

$$c = \frac{\sum_{i=1}^{N}\left[\frac{I_{exp}(s_i)I_{calc}(s_i)}{\sigma(s_i)^2}\right]}{\sum_{i=1}^{N}\left[\frac{I_{calc}(s_i)}{\sigma^2_{(s_i)}}\right]} \tag{5}$$

The structures ranking in the top 2% in both total energy and restraint energy were selected. To further narrow down the selection on the basis of the SAS data, we evaluated the $\chi^2$ distribution of the $^2$H-Nop5, $^2$H-L7Ae and $^2$H-RNA SANS curves. The SAS curves including the contribution from fibrillarin were left out, because we expected the position of fibrillarin to be variable when it is not in contact with the RNA. SAS fitness was calculated with the programs CRYSOL and CRYSON, from the ATSAS suite, version 2.7.5 (*Svergun et al., 1998*). Based on the distribution of fitness for all structures in each of the runs, we set loose cut-offs, which excluded only structures beyond the smooth, linearly increasing portion of the distribution curve. For the substrate D'-loaded complex, we selected structures within the top 90% ranking by $^2$H-RNA fitness $\chi^2 < 1.4\ \chi^2_{min}$ in the [off,off]-state, $\chi^2 < 2.2\ \chi^2_{min}$ in the [on,off]-state), the top 50% by $^2$H-L7Ae fitness ($\chi^2 < 1.3\ \chi^2_{min}$ in the [off, off]-state, $\chi^2 < 1.3\ \chi^2_{min}$ in the [on,off]-state), and the top 80% by $^2$H-Nop5 fitness ($\chi^2 < 6.1\ \chi^2_{min}$ in the [off,off]-state, $\chi^2 < 6.8\ \chi^2_{min}$ in the [on,off]-state); for the substrate D-loaded complex, we selected structures within the top 90% ranking by $^2$H-RNA fitness ($\chi^2 < 2.5\ \chi^2_{min}$ in the [off,off]-state, $\chi^2 < 3.4\ \chi^2_{min}$ in the [on,off]-state), the top 80% by $^2$H-L7Ae fitness ($\chi^2 < 2.0\ \chi^2_{min}$ in the [off,off]-state, $\chi^2 < 1.8\ \chi^2_{min}$ in the [on,off]-state) and the top 90% by $^2$H-Nop5 fitness ($\chi^2 < 6.5\ \chi^2_{min}$ in the [off,off]-state, $\chi^2 < 5.3\ \chi^2_{min}$ in the [on,off]-state). The average pair-wise RMSD of the structures of each ensemble, calculated over the C$\alpha$ and P atoms of the protein and RNA structured domains, including the fibrillarin units not bound to the RNA, was below 5 and 7 Å for the [on,off] and [off,off] conformers, respectively, with a maximum RMSD value of less than 10 Å in all cases.

Among the selected structures of each of the four runs ([on,off]- and [off,off]-states of both substrate D- and substrate D'-loaded sRNPs), the one with the lowest restraint-violation energy that maintained the correct RNA topology was chosen as the starting point for refinement in Cartesian space. The four refinement runs comprised 1500 structures each spanning up to 10 Å RMSD of C$\alpha$ and P atoms relative to the starting structure (number calculated for the substrate D'-loaded [on, off]-state). At the end of the refinement, we applied stringent selection criteria with respect to the SAS curves and loose criteria with respect to the energy. The cut-offs for the SAS data were set upon visual inspection of the $\chi^2$ distributions for each run and curve, whereby we allowed more structures to be selected when the $\chi^2$ distribution was flat.

For the substrate D'-loaded sRNP the cut-offs are as follows: top 33% of restraint-violation, van der Waals and total energy; top 83% for $^2$H-RNA ($\chi^2 < 1.3\ \chi^2_{min}$ for the [off,off]-state, $\chi^2 < 2.0\ \chi^2_{min}$ for the [on,off]-state); top 67% for $^2$H-L7Ae ($\chi^2 < 1.8\ \chi^2_{min}$ for the [off,off]-state, $\chi^2 < 1.1\ \chi^2_{min}$ for the [on,off]-state); top 33% for $^2$H-Nop5 ($\chi^2 < 2.7\ \chi^2_{min}$ for the [off,off]-state, $\chi^2 < 3.4\ \chi^2_{min}$ for the [on, off]-state); top 10% for $^2$H(70%)-Nop5-RNA ($\chi^2 < 6.1\ \chi^2_{min}$ for the [off,off]-state, $\chi^2 < 3.3\ \chi^2_{min}$ for the [on,off]-state). Applying these criteria we selected 1 structure for the substrate D'-loaded [off,off]-state and 12 structures for the [on,off]-state. The [on,off]-state structures displayed an average RMSD of 2.4 Å, calculated on all C$\alpha$ and P atoms (*Figure 3*) excluding the fully flexible regions, namely the free guide region of the RNA (nucleotides 51–62), the loops connecting the Nop5-NTD to the coiled-coil domain (residues 116–122), and the loops connecting the coiled-coil domain to the Nop5-CTD (residues 249–251).

For the substrate D-loaded sRNP the cut-offs are as follows: top 33% of restraint-violation, van der Waals and total energy; top 83% for $^2$H-RNA ($\chi^2 < 2.1\ \chi^2_{min}$ for the [off,off]-state, $\chi^2 < 2.2\ \chi^2_{min}$

for the [on,off]-state); 66% for $^2$H-L7Ae ($\chi^2$ < 2.7 $\chi^2_{min}$ for the [off,off]-state, $\chi^2$ <1.2 $\chi^2_{min}$ for the [on, off]-state); 33% for $^2$H-Nop5 ($\chi^2$ < 3.0 $\chi^2_{min}$ for the [off,off]-state, $\chi^2$ <1.9 $\chi^2_{min}$ for the [on,off]-state); 10% for $^2$H(70%)-Nop5-RNA ($\chi^2$ < 3.2 $\chi^2_{min}$ for the [off,off]-state, $\chi^2$ <2.8 $\chi^2_{min}$ for the [on,off]-state). The final ensembles for the substrate D-loaded [off,off]- and [on,off]-states consist of 3 and 20 structures, respectively, with a C$\alpha$ and P RMSDs of 4.6 and 2.4 Å, respectively.

Representative structures in the final ensembles were minimized in explicit water using Amber 14 and the corresponding Amber99SB force field (*Hornak et al., 2006*).

## Ensemble scoring

The PRE data and the SAS data indicated the presence of a conformational equilibrium between the [on,off]- and [off,off]-states, as discussed in the main text. The $^2$H-fibrillarin, $^2$H-fibrillarin/$^2$H-RNA SANS and SAXS curves were therefore fitted to a mixture of structures in the [on,off]- and [off,off]-states.

In order to address the flexibility of the Nop5-NTD–fibrillarin modules not in contact with the RNA, we sought to generate ensembles containing different orientations of these modules that would improve the fit to the SAS curves. This conformational diversity is in addition to the equilibrium between the [on,off]- and [off,off]-states, resulting in a pool of structures containing both [on, off]- and [off,off]- states and multiple conformations of Nop5-NTD–fibrillarin modules in each state.

To generate these ensembles we proceeded as follows. Starting from the representative structure of each ensemble of *Figure 3*, corresponding to the structure closest to the mean of the ensemble, we performed a further simulated-annealing step, where the loops connecting the Nop5-NTD–fibrillarin modules to the rest of the Box C/D particle were allowed to adopt random orientations, while the rest of the particle was kept rigid. At this stage, we generated 4000 structures with randomised Nop5-NTD–fibrillarin positions, from which we removed structures containing steric clashes. The structures also contained all spin-labels, which were left flexible, in order to allow back-calculation of PREs (see below).

In a separate run comprised of 300 structures, the template structures were kept entirely rigid while the spin-label side-chains were allowed to rotate in order to generate different orientations, as multiple conformations of the spin-label have been demonstrated to fit the PRE data more accurately than a single conformation (*Iwahara et al., 2004*).

Ensemble scoring was carried out for substrate D'- and substrate D-loaded sRNPs via the pseudo-genetic algorithm shown in *Figure 3—figure supplement 1b*. First, we grouped the structures into four pools, containing 3500, 3500, 300 and 300 structures: [on,off]-state with randomised Nop5-NTD−fibrillarin positions, [off,off]-state with randomised Nop5-NTD−fibrillarin positions, [on, off]-state with randomised spin-label orientations and [off,off]-state with randomised spin-label orientations. The algorithm generated four 'parent' ensembles, each comprising of 2–10 conformers randomly chosen from the pools. These ensembles were merged and sub-sampled, yielding 20 'children' sub-ensembles ranging from 3 to 10 conformers in size. Each sub-sampling event had a 30% probability of duplicating a conformer or replacing one with another from the main pool. The process of parent selection, sub-sampling and scoring was repeated 250 times.

The theoretical scattering curve of the ensemble was computed as the linear combination of the scattering curves of each individual conformer (scaling the populations to represent molar fractions rather than volume fractions, which is the standard ATSAS output). The $\chi^2$ value with respect to the experimental data was calculated by OLIGOMER (*Konarev et al., 2003*). The normalization of $\chi^2$ of all sub-sampled ensembles and across iterations was done according to *Equation 6* (*Karaca et al., 2017*):

$$\chi^2_{norm} = \frac{\chi^2_{ensemble} - \chi^2_{min}}{\chi^2_{max} - \chi^2_{min}} \tag{6}$$

where $\chi^2_{ensemble}$ is the fitness of an individual ensemble, and $\chi^2_{min}$ and $\chi^2_{max}$ are the respective minimum and maximum values across the iterations or sub-ensembles being considered. Five SAS curves were used for scoring: $^2$H-Nop5, $^2$H-Fib, $^2$H-Fib/$^2$H-RNA, $^2$H(70%)-Nop5/$^2$H-RNA and SAXS. The normalized $\chi^2$ values for each curve were then summed and renormalized into a single value, obtained with the same *Equation 6*, which then represented the overall SAS-fitness.

The calculation of the theoretical $I_{para}/I_{dia}$ ratios from mixed [on,off]- and [off,off]-state ensembles requires an estimation of the timescale of the exchange rate $k_{ex}$ ($k_{ex} = k_1 + k_{-1}$) between the [on,off]- and [off,off]-conformers. This can be easily done by inspecting the ILV-methyl $^1$H-$^{13}$C spectra of fibrillarin: in the case of slow conformational exchange, the methyl groups in the fibrillarin copy sampling the on- and off-states should each yield two separate NMR peaks, while for fast conformational exchange these methyl groups should each show only a single peak, at a position corresponding to the population-weighted average of the positions corresponding to the on- and off-states. To investigate this, we used the spectrum of the RNP assembled with ssR26 and loaded with substrate RNA as a reference for the slow-exchange situation: in this complex, two of the four fibrillarin copies adopt a stable on-state, while the other two are in the off-state, and a subset of the fibrillarin methyl groups show separate and resolvable peaks corresponding to the two states. In the spectra of the half-loaded st-sR26 RNPs we did not detect any peak at the positions corresponding to RNA-bound fibrillarin in the holo ssR26 RNP spectra, indicating that in the half-loaded mono-RNP, either the $k_{ex}$ is faster than the differences in the resonance frequencies of the fibrillarin methyl groups in the on- and off-states (~40–100 Hz), or the population of the on-state is too small to be detected. In the second case, one would expect no CSPs upon substrate RNA binding, which does not correspond with the observed spectra (*Figure 2a*, right panel), Thus, we back-calculated the PREs for the fibrillarin copy that can be in contact with the substrate–guide duplex using $<r^{-6}>$ensemble averaged distances over the [on,off]- and [off,off]-states, as appropriate for the fast exchange regime.

Each methyl group of each fibrillarin or L7Ae copy is influenced by two PRE tags (SL1 and SL2). The resulting $\Gamma_2$ values for the methyl groups of the two copies are given by:

$$\Gamma_2^{Methyl1} = \Gamma_{2,SL1}^{Methyl1} + \Gamma_{2,SL2}^{Methyl1}$$

$$\Gamma_2^{Methyl2} = \Gamma_{2,SL1}^{Methyl2} + \Gamma_{2,SL2}^{Methyl2} \tag{7}$$

where Methyl1 and Methyl2 refer to the two copies of L7Ae or fibrillarin. Because Methyl1 and Methyl2 have almost indistinguishable chemical shifts, the resulting $I_{para}/I_{dia}$ ratios for Methyl1 and Methyl2, calculated from *Equations (2) and (3)*, were averaged before comparison to the experimental data. The PRE fitness was quantified using $\chi^2$ to all experimental PRE values using *Equation (4)*. Distances were computed from the PDB files using the Biopython Bio.PDB module (*Cock et al., 2009*). The fitness of PRE data was normalized using *Equation (6)* and summed with the SAS-fitness score, to yield a consensus PRE-SAS score for each ensemble within the 20 sub-sampling events, and across the 250 iterations.

Three independent runs of the scoring algorithm were performed for substrate D'- and substrate D-loaded sRNPs, with the top scoring ensemble, judged by the consensus PRE-SAS score, displayed in *Figure 4*.

After this selection, the conformations of each individual tag were refined by generating additional 3000 conformers per tag and by using the same pseudo-genetic algorithm to select the ensembles of tag conformations that best fitted each individual PRE dataset. During this refinement step the positions of all proteins and RNA, as well as the populations of fibrillarin conformers in the ensemble, were left invariant, in order not to alter the fit to the SAS data.

## Molecular dynamics

Molecular dynamics simulations of the substrate D'- and substrate D-bound structures representing the [on,off]-states were carried out in AMBER 2018 (*Case et al., 2018*). The simulations were carried out in explicit TIP3P water using a cubic box with a 14 Å water layer and the ff14SB parameter set. The system was subjected to 20,000 cycles of solvent minimization with positional restraints on the complex (NPT), followed by heating to 328 K (NVT). The complete system was subjected to an additional 20,000 cycles of energy minimisation, and then allowed to relax, keeping restraints on the proteins and heavy atoms (NPT at 328 K, 0.5 ns). Subsequently, the two structures were subjected to a 150-ns molecular dynamics. Contacts were extracted using CPPTRAJ (*Roe and Cheatham, 2013*).

## Acknowledgements

The authors thank Dr. Artem Feoktystov (MLZ Munich) for assistance with recording and processing SANS data at KWS-1; Dr. Roman Lichtenecker (University of Vienna), for kindly providing the leucine methyl labelling precursor sodium [3-$^2$H$_2$,4-$^2$H, 5-$^{13}$C, 5'-$^2$ $_{H3}$]-α-ketoiso-caproate; Dr. Pawel Masiewicz (EMBL Heidelberg) and Susanne Zur Lage (HZI Braunschweig) for RNA production and Dr. Bernd Simon (EMBL Heidelberg) for assistance with structure calculations.

## Additional information

### Funding

| Funder | Grant reference number | Author |
|---|---|---|
| European Commission | FP7 ITN project RNPnet (contract number 289007 | Andrea Graziadei |
| Deutsche Forschungsgemeinschaft | CA294/3-2 | Teresa Carlomagno |

The funders had no role in study design, data collection and interpretation, or the decision to submit the work for publication.

### Author contributions

Andrea Graziadei, Resources, Data curation, Software, Formal analysis, Investigation, Visualization, Methodology; Frank Gabel, John Kirkpatrick, Data curation, Formal analysis; Teresa Carlomagno, Conceptualization, Data curation, Supervision, Funding acquisition

### Author ORCIDs

Andrea Graziadei (iD) https://orcid.org/0000-0001-7709-6002
Teresa Carlomagno (iD) https://orcid.org/0000-0002-2437-2760

### Decision letter and Author response

Decision letter https://doi.org/10.7554/eLife.50027.sa1
Author response https://doi.org/10.7554/eLife.50027.sa2

## Additional files

### Supplementary files

• Transparent reporting form

### Data availability

All data underlying the figures, including relevant structures, is available in Dryad with DOI https://doi.org/10.5061/dryad.q573n5tdv. All SAS curves have been deposited to the SASBDB under accession codes SASDGV2, SASDGW2, SASDGX2, SASDGY2, SASDGZ2, SASDG23, SASDG33, SASDG43, SASDG53, SASDG63, SASDG73, SASDG83, SASDG93, SASDGA3 (project accession code 860).

The following datasets were generated:

| Author(s) | Year | Dataset title | Dataset URL | Database and Identifier |
|---|---|---|---|---|
| Graziadei A, Gabel F, Kirkpatrick J, Carlomagno T | 2020 | Data from: The guide sRNA sequence determines the activity level of Box C/D RNPs | https://doi.org/10.5061/dryad.q573n5tdv | Dryad Digital Repository, 10.5061/dryad.q573n5tdv |
| Graziadei A, Gabel F, Kirkpatrick J, Carlomagno T | 2019 | SAS data from: The guide sRNA sequence determines the activity level of Box C/D RNPs | https://www.sasbdb.org/project/860/ | Small Angle Scattering Biological Data Bank, project 860 |

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
