## [Decision Letter]

**Acceptance summary:**

A relatively large macromolecular RNA protein complex, Box C/D RNP, was known to methylate ribosomal RNA at ribose 2'-OH, but how this activity is regulated has been poorly understood. The authors show here that the substrate RNA sequence outside of the methylation site can impact C/D RNP mediated interactions and rRNA methylation. This work provides a strong example of the power of using multiple biophysical methods in addressing important structural biology problems, as well as a compelling example of the role of NMR in such studies.

**Decision letter after peer review:**

Thank you for submitting your manuscript, "The guide sRNA sequence determines the activity level of Box C/D RNPs" for consideration in *eLife*. Your manuscript has been reviewed by two experts and both they and I are impressed with the range of advanced methodologies that has been used to study such a large complex by NMR. We also feel that this can be an important contribution should you and your colleagues address the concerns listed below. Please note that further extensive experiments are not required, but that conclusions must be strongly supported by experiment.

The reviewers, which have opted to remain anonymous, have discussed the reviews with one another, and the Reviewing Editor has drafted this decision to help you prepare a revised submission.

Major

1) Previous work by the group described a di-RNP complex in the presence of substrate (holo di-RNP); this work describes a mono-RNP complex in the presence of substrate (holo mono-RNP). The authors mention this but don't provide details. The questions that arise are: a) why are the previous and the present complexes different, is there a specific RNA contact that drives the formation of the di-RNP? and b) in case small differences in the guide RNA have a pronounced effect on the fully assembled complex, what stoichiometry do the authors expect for the Box C/D complex with the native guide RNA?

2) The substrate RNA that the authors use is significantly shorter than the natural substrate RNA (rRNA is long). Are the presented structures compatible with much longer substrate RNAs and is the proposed mechanism between the methylation efficiency of D' and D substrates also valid in the context of longer substrates? Can the terminal bases of the 3' and 5' ends of the substrate RNA that is used here adopt the same conformation when extra nucleotides are present?

3) The authors should be more accurate regarding the description of the RNA that they used. In the text and Materials and methods, the authors write that they use a stabilized version of the sRNA (for what reason, this is not explained, see also remark 1). In addition, Figure 1B displays two sRNA sequences (the WT and the stabilized one), the legend refers to this as "Sequence of the sRNAs used…" We assume the authors mean that the lower one of the two is the used sRNA sequence. Likewise, Figure 2 refers to a ssR26 species. We assume that this is the ss-sR26 RNA, the RNA that was used in a previous study. Such inaccuracies often cause confusion regarding the type of the RNA that is used. One suggestion is to add this info to the cartoons in Figure 1A, where cartoon of the di-RNP-apo, mono-RNP-holo and di-RNP-holo complexes are shown.

4) "In the di-RNP complex, the methyl-group NMR spectra indicated the presence of fibrillarin in two states, one close to the substrate-guide duplex (on-state) and one far from the substrate (off-state) (Figure 2A)." We don't see this. In Figure 2A, the NMR spectra of the complex assembled with ss-sR26 (previous study; cyan) and the complex assembled with st-sR26 (this study, blue) are overlaid. According to SAXS data these are both di-RNPs. However, we don't see any sign of the presence of two states in those NMR spectra. In addition, later in the manuscript, the authors conclude that the RNP is dynamic in the presence of substrate, whereas the displayed spectra are from complexes in the absence of substrate.

5) "In contrast, both half-loaded mono-RNPs (substrate D- bound and substrate D'-bound) show only one set of fibrillarin resonances; however, the peaks that are split in two in the di-RNP are noticeably elongated in the mono- RNP, suggesting a conformational equilibrium between on- and off-states interchanging at a rate that is fast compared to the corresponding difference in NMR frequencies." Again, we are not following this:

a) In Figure 2B, we see one resonance for the st-sR26 apo complex (blue, same spectrum as panel a. This contradicts with statement above, that the apo-di RNP samples two states.

b) For the st-sR26 holo complex (green, mono-RNP holo) (is this after addition of both D and D' or after either D or D'?), we see one maybe slightly broadened resonance. For the ssR26-holo (purple, di-RNP holo) complex one can imagine two resonances (but the ssR26 RNA is not really the subject to this study). This could suggest that the holo complexes (mono- or di-RNA) are actually sampling multiple states. The text, however, mentions that the half-loaded mono-RNPs only show one set of resonances.

6) We also don't fully understand the statement in the legend of Figure 2B "a single but elongated peak, characteristic of one copy of fibrillarin being in fast-exchange between the two states", refering to the green contours from the mono-RNP holo complex. In case there is fast exchange between 2 states, the green resonance should appear between the two purple resonances. In our view, this is not the case; the broad resonance appears exactly at one of the two signals that form the purple "doublet". Also, the broadening of the green resonance appears to be mainly in the carbon dimension, and not in the proton dimension. Are the spectra recorded with exactly the same experimental settings, or is it possible that differences in processing, viscosity, temperature or acquisition time is the cause of the differential linewidth? In summary, the conclusion regarding the dynamics of the complex seems not supported by the displayed NMR spectra.

7) Does Figure 2C belong to the st-sR26 holo complex in Figure 2B? If yes, this should be clearly indicated.

8) Figure 2D, top. How can I_para_/I_dia_ be (significantly) larger than 1? For readability, please label in the figure that the blue curve is for D' bound and that the red one is for D bound and indicate that this is the mono-RNP holo complex.

9) Do the I_para_/I_dia_ values depend on the excess of the D or D' RNA? Are the binding sites for D and D' in both cases fully saturated? We assume that less saturation results in more mobility, so it is important to ensure that one compares fully saturated complexes.

10) For Figure 2D, bottom, the deviations between the experimental and simulated P(r) values are similar for the short distances (0 – 40 A) and for the large distances (70 – 140 A). The deviations for the small r values are independent of conformational changes and these are thus intrinsic to the method. The question is, if the deviations at larger r values are thus significant enough that one can conclude from these SANS data that there are two states? If yes, does this agree with the population ratio of the [on, off] and [off, off] states that the authors determined later in the manuscript?

11) Subsection “The conformational ensembles of the half-loaded mono-RNPs in solution” paragraph three: "show a reasonable similarity". The authors need to provide at least an RMSD and an overlay of their two structures with the structure of Lin et al. to support this statement.

12) In context of sRNP complex, the affinity of fibrillarin for substrate depends on RNA sequence beyond recognition site

– It is unclear what the recognition site is. Referring to substrate sequences? Without a clearly defined recognition site, this claim is not supported by the data

– Do the authors mean methylation site rather than recognition site as specified in the Discussion? If this is the case, the Abstract/Introduction should be modified to reflect this level of detail.

[Editors' note: further revisions were suggested prior to acceptance, as described below.]

Thank you for submission of a revised version of your manuscript "The guide sRNA sequence determines the activity level of Box C/D RNPs" to e*Life*. The paper has been re-evaluated by the reviewers. The reviewers feel that the manuscript is much improved over the initial submission but bring up a number of critical points that must be addressed properly should the manuscript be accepted for publication.

In short, there are serious questions about the NMR data. For example, it is extremely difficult to establish that a dynamic equilibrium is occurring between on/off conformations for one of the copies of fibrillarin from the CSP data. Second, it would be useful to show that the PRE data are inconsistent with the off state and presumably also with the on state. The authors attempt to say this and refer to Figure 2C but it is not possible to see much from this figure, as it is small. It would be important to show the structures of the on and off state along with the PRE values and establish that the PRE values can simply not be explained by a static entity. We suggest separating the on and off structures pictorially to make the point. Second, given that some of the PREs are > 1 it is important that they be validated. One way of doing this is to establish that intra-protein PREs (i.e., PREs from probes attached to the same protein as spin label) are consistent with expectations based on distances.

Major comments

Previous points #4, 5 and 6; Legend Figure 2B: "Despite the absence of peak splitting, peak budding is visible in the spectrum of the substrate D'-loaded st-sR26 RNP in all instances for which peak splitting or budding is observed in the spectrum of the holo ssR26 di-RNP."

1) This figure and its legend have been remade for this version of the manuscript. Nevertheless, we cannot follow what the authors see in their spectra. The only instance, where we can see a peak splitting (not sure if "budding" is a term that can be used), is in the purple (ssR26 holo) spectrum for residue L58 and, with some imagination, for I62. For all other panels and all other complexes, we fail to see any change in the resonance. In other words, for V118, L160, V141 and L200, there are effectively no chemical shift perturbations that are larger than 0.01 ppm in proton. In carbon there is no CSP whatsoever. For the open-closed conformations, one would expect many and large CSPs.

This is a very important point, as the authors have little convincing NMR data that show that the complex exists in two states with fibrillarin either in the on- or the off-state.

2) Previous point #8; Figure 2C. The PRE data with an I_para_/I_dia_ still are very problematic. It is physically not possible that this ratio is larger than 1. In the plot, ratios that are 1.25 are shown. The authors write in their letter that there are only two values over 1, in the plot there appear to be at least 6. The remark that it is hard to extract reliable data for crowded spectral regions is correct, however, in case the data are unreliable or highly uncertain (which is not reflected in the presented error bars), it is not possible to draw solid conclusions. So how do the authors come to their conclusions based on the presented data? As an example, a ratio of 0.75 is mentioned in the text as highly relevant for the closed conformation, whereas a ratio of 1.25 is rebutted (only in the letter, not in the manuscript) as resulting from difficult spectral interpretations.

As the NMR data do not agree with SANS data, the authors involve conformational rearrangements between open and closed conformations. Clearly, in case things are not in agreement, it can always be explained by dynamics. Because of the very weak NMR evidence (2 points above), it seems a far stretch to conclude that the complex undergoing open-closed motions.

Previous point #10; Figure 2D. The figure is still hard to understand. The substrate D loaded experimental (red) and theoretical (brown) are very different. The substrate D' loaded experimental (cyan) and theoretical (blue) are also very different.

[Editors' note: further revisions were suggested prior to acceptance, as described below.]

Thank you for resubmitting your work entitled "The guide sRNA sequence determines the activity level of Box C/D RNPs" for further consideration by *eLife*. Your revised article has been evaluated by Philip Cole (Senior Editor) and a Reviewing Editor.

Thank you for your letter addressing the comments that were raised based on the second revision and you are invited to resubmit a revised manuscript. Let me first address whether it is worth submitting a further revised paper. I agree that it is a lot of work on all sides – but there was very significant enthusiasm from the reviewers and me, and I think that we would all like to see some final issues clarified.

First, when there are issues raised by the reviewers that seem unreasonable to the authors, this sometimes reflects the fact that the material was not explained as clearly as might be possible. One of the advantages of *eLife* is that space is not nearly as constrained as in some other prestigious journals. I would suggest taking advantage of this. For example, include validation of the PREs (intra) for both fibrillarin and L7Ae, as a main figure, to convince the reader that your data are good (I believe that this will placate the reviewers). Then if you fit your inter-data to 1 structural model you should be able to show that the fit is worse. We request that you do this, as then you establish that there could be more than 1 structure. We request that you also include structural models of the sort in Figure 2C but larger so that these can be better viewed. Following this up with the SAXS data will then make your thought process transparent. As the reviewers had difficulty with the SAXS please make it very clear that the disagreement is a really powerful argument for why there are dynamics.

We suggest that the NMR CSPs are less convincing because they are small. We recommend that you might emphasize this before the PREs and SAXS (your ordering seems very logical to me) but tone it down a bit as the differences in shifts are not large. One can sort of imagine that the green peaks are between blue and purple but we are talking about very small changes. If you say this as a follow-in to the other data, we think that it may sit easier with the reviewers.

Finally, regarding the errors in PREs. we can appreciate that some values could well be > 1 given statistics. One would hope that the errors would reflect this. For a couple of cases that we count, this is not the case. Perhaps if you show more PRE data this will help – and as you will hopefully be showing correlation plots assuming 1 state and that these are not as good as the intra-validation – showing more raw data would be justified.

---

## [Author Response]

Major1) Previous work by the group described a di-RNP complex in the presence of substrate (holo di-RNP); this work describes a mono-RNP complex in the presence of substrate (holo mono-RNP). The authors mention this but don't provide details. The questions that arise are: a) why are the previous and the present complexes different, is there a specific RNA contact that drives the formation of the di-RNP? and b) in case small differences in the guide RNA have a pronounced effect on the fully assembled complex, what stoichiometry do the authors expect for the Box C/D complex with the native guide RNA?

We thank the reviewers for this question. We have thoroughly investigated this point during the course of this work, as the discovery that the complexes assembled with the sR26 and st-sR26 RNAs form monomeric species upon substrate binding was unexpected.

With regard to point b): RNPs assembled with both sR26 and st-sR26 RNAs transition from the di-RNP to the mono-RNP form upon substrate binding. To demonstrate this, we have added Figure 1—figure supplement 4, which shows the SAXS curves of apo and holo-complexes assembled with sR26.

With regard to point a): our experiments revealed that, while the apo complex is always a di-RNP, the oligomerization state of the holo complex depends on the sequence of the guide RNA in the substrate recognition regions. In particular, the sequences directly upstream of either box D or box D’ play a fundamental role (see new Figure 1—figure supplement 5). Unfortunately, because of the lack of high-resolution data for the disordered parts of the RNA components of the complex, we do not yet know why this is the case and cannot propose a predictive set of rules. We are planning solid-state NMR experiments to answer this question. However, these experiments have never been tried before and are likely to require substantial experimental effort.

Following the question of the reviewers we have now added new experimental data (Figure 1—figure supplement 5) and an entire new paragraph to address this point.

“Before embarking upon the structural study of the sRNPs containing st-sR26, we first wanted to understand which elements are responsible for the different oligomerization states of the holo ssR26- and holo st-sR26-RNPs. The ssR26 and the st-sR26 RNAs differ only in the sequence of the guide RNA at the box D position, which in the case of ssR26 is identical to that of guide D’. Thus, we generated two additional guide RNAs with distinct D and D’ sequences, st-sR26-1 and st-sR26-2: in st-sR26-1 (st-sR26-2), guide sequence D is a chimeric sequence, formed by the 5’ half of st-sR26 guide D (st-sR26 guide D’) and the 3’ half of st-sR26 guide D’ (st-sR26 guide D) (Figure 1—figure supplement 5A). Interestingly, the Box C/D enzyme containing st-sR26-1 maintained the di-RNP architecture upon binding of either substrate RNAs, while the sRNP comprising st-sR26-2 transitioned to the mono-RNP state (Figure 1—figure supplement 5B). Mutation of the last nucleotide of st-sR26-1 guide D to either C or U (A61C and A61U with complementary substrate D) did not perturb the di-RNP architecture (Figure 1—figure supplement 5C). We conclude that the guide sequence strongly influences the oligomerization state of the holo complex. “

2) The substrate RNA that the authors use is significantly shorter than the natural substrate RNA (rRNA is long). Are the presented structures compatible with much longer substrate RNAs and is the proposed mechanism between the methylation efficiency of D' and D substrates also valid in the context of longer substrates? Can the terminal bases of the 3' and 5' ends of the substrate RNA that is used here adopt the same conformation when extra nucleotides are present?

Both the 3’ and 5’ ends of the substrates used in this work are not involved in base-pairing and can be easily elongated without perturbing the structure of the Box C/D RNP. We use substrates that are longerthan the 11 base-pairing nucleotides, in order to account for the longer natural substrates.

There are several reports in the literature stating that longer RNAs are methylated more efficiently in vitro, probably due to additional interactions within the Box C/D RNP, which however remain uncharacterized. rRNA methylation partially occurs during transcription; thus, another question one can ask is whether the sequences flanking the methylation regions at more than 5–6 nucleotides distance are involved in interactions with other proteins or other RNA stretches and thus unavailable for additional interactions within the Box C/D RNP. We do not know the answer to these questions. There is currently no good model to test the effect of the longer rRNA sequences in vivo.

3) The authors should be more accurate regarding the description of the RNA that they used. In the text and Materials and methods, the authors write that they use a stabilized version of the sRNA (for what reason, this is not explained, see also remark 1). In addition, Figure 1B displays two sRNA sequences (the WT and the stabilized one), the legend refers to this as "Sequence of the sRNAs used…" We assume the authors mean that the lower one of the two is the used sRNA sequence. Likewise, Figure 2 refers to a ssR26 species. We assume that this is the ss-sR26 RNA, the RNA that was used in a previous study. Such inaccuracies often cause confusion regarding the type of the RNA that is used. One suggestion is to add this info to the cartoons in Figure 1A, where cartoon of the di-RNP-apo, mono-RNP-holo and di-RNP-holo complexes are shown.

We apologize for this. We have now added the ssR26 RNA sequence to panel B of Figure 1. In addition, we have changed the figure caption to:

“Two RNA sequences (st-sR26 and ssR26) were derived from the Pf sR26 RNA and used to assemble the Box C/D sRNPs either in this (st-sR26) or previous studies (ssR26, (Lapinaite et al., 2013)). The sequence of st-sR26 is derived from the native sR26 RNA upon substitution of the apical K-loop element with the more stable K-turn element. This substitution does not affect the oligomerization state of the complex, as shown in Figure 1—figure supplement 4, but ensures the stability of the complex over several days.”

This text explains why we used the st-sR26 RNA instead of sR26 and also addresses the question raised by the reviewers with respect to the oligomerization state expected for the complex assembled with the sR26 RNA.

4) "In the di-RNP complex, the methyl-group NMR spectra indicated the presence of fibrillarin in two states, one close to the substrate-guide duplex (on-state) and one far from the substrate (off-state) (Figure 2A)." We don't see this. In Figure 2A, the NMR spectra of the complex assembled with ss-sR26 (previous study; cyan) and the complex assembled with st-sR26 (this study, blue) are overlaid. According to SAXS data these are both di-RNPs. However, we don't see any sign of the presence of two states in those NMR spectra. In addition, later in the manuscript, the authors conclude that the RNP is dynamic in the presence of substrate, whereas the displayed spectra are from complexes in the absence of substrate.

We apologize for having generated confusion with our typo. The sentence should have made referenceto Figure 2B and not 2A. As the reviewers and the editor correctly noticed, there is no evidence of multiple fibrillarin states in the apo form of the complexes. The apo complexes assembled with either ssR26 or st-sR26 are both di-RNPs, with all four copies of fibrillarin far from the RNA due to the absence of the substrate RNAs. The two spectra of Figure 2A (now Figure 2A, left panel) overlap well with each other and indeed are not expected to differ. We have corrected the typo, re-designed Figure 2, substantially changed the main text and the figure caption to provide answers to this as well as to the next points. With respect to this point, the figure caption now reads:

“Overlay of ILV-methyl ^1^H-^13^C spectra of fibrillarin in the apo ssR26 (turquoise) and apo st-sR26 (blue) RNPs. In both di-RNPs, all four fibrillarin copies are distant from the RNA and the two spectra are identical.”

5) "In contrast, both half-loaded mono-RNPs (substrate D- bound and substrate D'-bound) show only one set of fibrillarin resonances; however, the peaks that are split in two in the di-RNP are noticeably elongated in the mono- RNP, suggesting a conformational equilibrium between on- and off-states interchanging at a rate that is fast compared to the corresponding difference in NMR frequencies." Again, we are not following this:a) In Figure 2B, we see one resonance for the st-sR26 apo complex (blue, same spectrum as panel a. This contradicts with statement above, that the apo-di RNP samples two states.

Please see the answer to the point above.

b) For the st-sR26 holo complex (green, mono-RNP holo) (is this after addition of both D and D' or after either D or D'?), we see one maybe slightly broadened resonance. For the ssR26-holo (purple, di-RNP holo) complex one can imagine two resonances (but the ssR26 RNA is not really the subject to this study). This could suggest that the holo complexes (mono- or di-RNA) are actually sampling multiple states. The text, however, mentions that the half-loaded mono-RNPs only show one set of resonances.

The green spectrum of Figure 2B referred to the fully loaded st-sR26 RNP, which has a complicated behavior, as both fibrillarin copies are in equilibrium between the RNA-bound and the RNA-free states. The relative populations of the RNA-bound forms of the fibrillarin at the substrate-loaded site are different for the two different half-loaded mono-RNPs; in addition, the chemical shifts of the respective RNA-bound states will also show differences, as the substrates have different sequences (D and D’). By showing the holo form of the st-sR26 complex we have generated much confusion, especially as we did not explain the spectrum. In this revised version we have re-designed the representation of the spectra in Figure 2: we now show the spectrum of the substrate D’-loaded mono-RNP, which is directly comparable to that of the holo ssR26 di-RNP, as the substrate sequences are the same. We also show the expanded view of more methyl group peaks to show that peak “budding” effects are present in both the ^1^H and the ^13^C dimensions and always appear in the same direction of the peak splitting or budding observed in the holo ssR26 di-RNP.

We have also added two paragraphs in the main text to explain our interpretation of the peak shape. However, we would like to stress that the presence of the conformational equilibrium is not proven by the minuscule peak budding in the HMQC spectra, but rather by the incompatibility of the SANS and PRE data. Thus, by comparing the HMQC spectra, we do not want to prove the existence of the conformational equilibrium, but rather verify that the peak shapes are compatible with the presence of this equilibrium. Here, we summarize the explanation that we give in the text:

1) In the apo complexes all fibrillarin copies are distant from the RNA in both the st-sR26 and ssR26 RNPs. The spectra of the two species overlap perfectly, as shown in Figure 2A, left.

2) In the holo di-RNP complex assembled with the ssR26 RNA (our previous work), all structural data are compatible with two fibrillarin copies being in stable contact with the RNA and two being far from the RNA. Two distinct NMR peaks are observed for some of the fibrillarin methyl groups, while many other peaks showed substantial asymmetric broadening (“budding”). Thus, the position of the second peak in the di-RNP HMQC spectrum is representative of fibrillarin bound to the substrate D’–guide duplex, as all substrate-binding sites in this di-RNP recognize substrate D’.

3) In the mono-RNP half-loaded with substrate D’, the spectra appear to show only one set of peaks. However, the peaks are elongated (“budding”) in the direction of the second peak seen in the di-RNP spectra. The half-loaded mono-RNP bound to substrate D’ contains two fibrillarin copies. One fibrillarin copy never binds the RNA, as its corresponding substrate (in this case substrate D) is absent. Thus, its methyl group peaks are expected to appear at the same positions as in the apo complex. The other fibrillarin copy may bind the RNA. If this fibrillarin copy were stably bound to the RNA, we would see peaks at a similar position as the second peaks in the holo ssR26 di-RNP spectrum, but this is not the case. If this fibrillarin copy were rapidly exchanging between RNA-bound and -unbound forms, we should see a peak halfway between the two peaks seen in the holo ssR26 di-RNP HMQC spectrum, assuming that the population distribution is 50%:50% and that the line-widths of the two states are similar. This peak, however, would overlap strongly with the peak of the first fibrillarin copy, which stays in the unbound form, because of the modest chemical shift differences between the RNA-bound and RNA-unbound states (see spectra of the di-RNP) as well as the large line-widths. Thus, the observed peak-shape originates from the overlap of the peak of one exchanging fibrillarin molecule with the peak of one non-exchanging fibrillarin molecule in the RNA-unbound state. We have now provided more examples of peaks in Figure 2B and simplified the spectral overlay to allow the reader to appreciate this.

6) We also don't fully understand the statement in the legend of Figure 2B "a single but elongated peak, characteristic of one copy of fibrillarin being in fast-exchange between the two states", refering to the green contours from the mono-RNP holo complex. In case there is fast exchange between 2 states, the green resonance should appear between the two purple resonances. In our view, this is not the case; the broad resonance appears exactly at one of the two signals that form the purple "doublet". Also, the broadening of the green resonance appears to be mainly in the carbon dimension, and not in the proton dimension. Are the spectra recorded with exactly the same experimental settings, or is it possible that differences in processing, viscosity, temperature or acquisition time is the cause of the differential linewidth? In summary, the conclusion regarding the dynamics of the complex seems not supported by the displayed NMR spectra.

Please see the answer to the question above, as well as the new Figure 2. The spectra of the apo and substrate D’-loaded sRNPs were acquired and processed in exactly the same way (as well as all other spectra). The display of additional peaks in Figure 2B now demonstrates that the “budding” does not occur exclusively in the ^13^C dimension. In addition, the clearer display of the peaks should allow better appreciation of the differences in the line-shapes of the peaks of the apo versus the substrate D’-loaded st-sR26 RNPs, which are consistent with (but do not prove) the conformational equilibrium.

The difference between “budding” peaks (those belonging to methyl groups which are split or budding in the holo di-RNP) and non-budding peaks (those belonging to methyl groups which are NOT split or budding in the holo di-RNP) can be appreciated from the lower leftmost panel of Figure 2B, where L160 (budding in the holo ssR26 di-RNP, purple spectrum) shows budding upon addition of substrate D’ to the st-sR26 RNP, while L66 does not. This comparison also demonstrates that the budding is not an artifact generated by different experimental or processing parameters. We have now commented on this in the figure caption.

7) Does Figure 2C belong to the st-sR26 holo complex in Figure 2B? If yes, this should be clearly indicated.

We have amended the figure and figure caption accordingly.

8) Figure 2D, top. How can I_para_/I_dia_ be (significantly) larger than 1? For readability, please label in the figure that the blue curve is for D' bound and that the red one is for D bound and indicate that this is the mono-RNP holo complex.

Only two values of I_para_/I_dia_ are larger than 1 (between 1.1 and 1.2). We often encounter these cases when quantifying PREs, due to errors in line-shape fitting as a consequence of spectral overlap and due to noise. The quantification of the I_para_/I_dia_ values in regions of spectral overlap may be difficult. To address this problem, we exclude the more overlapped peaks from the analysis and we apply a generous error-range on the extracted distances (lower-bound of 2 Å, independent of the noise-derived error).

We have amended the figure according to the reviewers’ suggestion.

9) Do the I_para_/I_dia_ values depend on the excess of the D or D' RNA? Are the binding sites for D and D' in both cases fully saturated? We assume that less saturation results in more mobility, so it is important to ensure that one compares fully saturated complexes.

The I_para_/I_dia_ ratios do depend on the saturation of the D and D’ sites as they are different for the apo and holo states and for the half-loaded substrate D- and substrate D’-bound mono-RNPs. To verify that both sites are saturated we measured the 1D ^1^H spectra of the sRNP with increasing concentrations of substrate and followed the appearance of free substrate in the spectral region of the RNA H1’ peaks, which is devoid of protein peaks. We see no free-RNA peaks for a sub-stoichiometric ratio of substrate RNA:enzyme (see spectrum in Author response image 1), but we see free RNA peaks (sharp peaks) at a molar excess of 1.25:1 (starting from 1:1 due to small errors in calculating the concentration of the enzyme effectively present in solution). This demonstrates that at a stoichiometric ratio of 1:1 the guide RNA binding site is completely saturated with substrate RNA. To be on the safe side, we use a 1.2 molar excess of substrate RNA in all experiments.

We have now added a sentence in the Experimental Section to clarify this point.

**Author response image 1. respfig1:** Titration of substrate RNA onto the apo Box C/D sRNP.

10) For Figure 2D, bottom, the deviations between the experimental and simulated P(r) values are similar for the short distances (0 – 40 A) and for the large distances (70 – 140 A). The deviations for the small r values are independent of conformational changes and these are thus intrinsic to the method. The question is, if the deviations at larger r values are thus significant enough that one can conclude from these SANS data that there are two states? If yes, does this agree with the population ratio of the [on, off] and [off, off] states that the authors determined later in the manuscript?

When we fit SAS data, we directly fit the scattering curves rather than the P(r) distributions. If we calculate the structures of the half-loaded sRNPs assuming only one state of each half-loaded complex, namely the [on,off]-state as required by the PRE data, the resulting structures do not fit the directly measured SANS and SAXS curves. This is the main argument to support the existence of a conformational equilibrium. We show this now in Figure 2D and in Figure 2—figure supplement 3.

The choice of displaying the P(r) distributions in Figure 2D rather than the direct SAS scattering curves is done for visual reasons, as the P(r) distribution function is easier to interpret intuitively. The discrepancy between the experimental and theoretical functions noted by the reviewers in the first version of this manuscript was due to the fact that in the old Figure 2D, instead of calculating the P(r) curves from the simulated SANS curves of the structural ensembles, we displayed the histogram of the pair-wise Cα–Cα and P–P distances scaled by the respective scattering contrast of ^2^H-protein and ^1^H-RNA in 42%:58% D_2_O:H_2_O solvent mixture. This calculation does not take into consideration the fact that the scattering of the ^1^H-proteins is only perfectly matched at the beginning of the SANS curve and that atoms other than the Cα or P atoms contribute to the scattering. In the new version of Figure 2D we calculate the theoretical P(r) functions from the theoretical ^2^H-Fib SANS curves averaged over the ensemble of [on,off]-structures that satisfy the PRE data of either the substrate D- or substrate D’-loaded sRNPs. Now, the theoretical P(r) functions at short distances perfectly match the experimental ones (as at short distances the P(r) function represents the pairwise distance distribution of atoms in one fibrillarin copy), while they differ at large distances demonstrating the incompatibility of the SANS data with a pool of [on,off]-structures only.

11) Subsection “The conformational ensembles of the half-loaded mono-RNPs in solution” paragraph three: "show a reasonable similarity". The authors need to provide at least an RMSD and an overlay of their two structures with the structure of Lin et al. to support this statement.

The overlays are shown in Figure 3—figure supplement 2. We have now added the RMSDs to the figure caption.

12) In context of sRNP complex, the affinity of fibrillarin for substrate depends on RNA sequence beyond recognition site– It is unclear what the recognition site is. Referring to substrate sequences? Without a clearly defined recognition site, this claim is not supported by the data– Do the authors mean methylation site rather than recognition site as specified in the Discussion? If this is the case, the Abstract/Introduction should be modified to reflect this level of detail.

We have changed “recognition site” to “methylation site” as this is what we mean.

[Editors' note: further revisions were suggested prior to acceptance, as described below.]

Major commentsPrevious points #4, 5 and 6; Legend Figure 2B: "Despite the absence of peak splitting, peak budding is visible in the spectrum of the substrate D'-loaded st-sR26 RNP in all instances for which peak splitting or budding is observed in the spectrum of the holo ssR26 di-RNP."1) This figure and its legend have been remade for this version of the manuscript. Nevertheless, we cannot follow what the authors see in their spectra. The only instance, where we can see a peak splitting (not sure if "budding" is a term that can be used), is in the purple (ssR26 holo) spectrum for residue L58 and, with some imagination, for I62. For all other panels and all other complexes, we fail to see any change in the resonance. In other words, for V118, L160, V141 and L200, there are effectively no chemical shift perturbations that are larger than 0.01 ppm in proton. In carbon there is no CSP whatsoever. For the open-closed conformations, one would expect many and large CSPs.This is a very important point, as the authors have little convincing NMR data that show that the complex exists in two states with fibrillarin either in the on- or the off-state.

As noted also in the previous answers, it is not the NMR data by itself that indicates the existence of the conformational exchange but the combination of PRE and SAS data. To make this absolutely clear we have rewritten the entire Results section and remade Figure 2. We no longer discuss chemical shifts in terms of conformational exchange but focus only on the discrepancy between PRE and SAS data when attempting to fit them to a single conformational state. To provide further evidence that the conformational exchange is needed to fit the data we now compare the PRE and SAS fits for: the representative structures of [on,off] and [off,off]-states (Figure 3—figure supplements 3, 4 and 5); ensembles of mixed [on,off]- and [off,off]-conformers (Figure 4 and new Figure 5); ensembles of [on,off]- OR [off,off]-conformers only (Figure 4—figure supplement 1 and Figure 5—figure supplement 1). While the PRE data can be fit in multiple scenarios, structural ensembles containing both [on,off]- and [off,off]-conformers (Figure 4 and new Figure 5) can properly fit the PRE and SAS data simultaneously. We are confident that this line of reasoning together with the new analysis presented in the supplements to Figure 3, supplement to Figure 4, new Figure 5 and its supplement, should be convincing.

Please also note that we do not agree with the sentence “For the open-closed conformations, one would expect many and large CSPs.” Fibrillarin interacts with the RNA backbone and this interaction is driven by arginine and lysine amino acids, as well as other polar side-chains, but definitely not by methyl-group-containing amino acids. Indeed, of all the fibrillarin ILV methyl groups, only those of V110 are expected to be within 5 Å of the RNA. Thus, we expect only a few detectable methyl-group CSPs upon RNA binding. Nonetheless, to demonstrate more clearly that there are indeed CSPs upon RNA binding, we now show a different region of the spectrum containing peaks from the residues expected to be closest to the RNA. This region was not shown in the original version of the paper as the peaks are quite overlapped. Nevertheless, the CSPs are clear, although difficult to quantify due to the overlap.

2) Previous point #8; Figure 2C. The PRE data with an I_para_/I_dia_ still are very problematic. It is physically not possible that this ratio is larger than 1. In the plot, ratios that are 1.25 are shown. The authors write in their letter that there are only two values over 1, in the plot there appear to be at least 6. The remark that it is hard to extract reliable data for crowded spectral regions is correct, however, in case the data are unreliable or highly uncertain (which is not reflected in the presented error bars), it is not possible to draw solid conclusions. So how do the authors come to their conclusions based on the presented data? As an example, a ratio of 0.75 is mentioned in the text as highly relevant for the closed conformation, whereas a ratio of 1.25 is rebutted (only in the letter, not in the manuscript) as resulting from difficult spectral interpretations.

From the original PRE publication of Battiste and Wagner (Battiste and Wagner, 2000) onwards, all I_para_/I_dia_ data-sets reported in the literature contain values higher than 1 and even 1.1:

**Author response image 2. respfig2:** Examples of experimental PRE intensity-ratios in the literature.

Excerpt from the text of Battiste and Wagner: “The apparent noise or variation of intensity ratios for cross-peaks that should be far away from the spin-label nitroxide (no broadening effect) is approximately 10-15%. “

Other examples of PRE values higher than 1.1 can be found in Huang et al. Scientific Reports 6, Article number: 33690 (2016, Sjodt and Clubb, Bio Protoc. 2017 Apr 5; 7(7): e2207and many other publications.

Nevertheless, in order to address the concern regarding the values higher than 1, we have reprocessed all spectra to correct for a slight baseline offset present in three of them. We have then extracted all PREs from the newly processed spectra and confirmed that they are very similar to the old values. In addition we have increased the errors on the I_para_/I_dia_ values to at least 10%, as recommended by Battiste and Wagner, to take into account possible changes in intensity due to sample manipulation and/or sample ageing.

For each of the D’-loaded and D-loaded complexes, we have ~15 values of (I_para_/I_dia_- error) larger than 1.1 among a total of 449 and 414 PREs, respectively. These numbers correspond to ~6% of the 258 (D’-loaded) and 261 (D-loaded) I_para_/I_dia_ ratios higher than 0.9, namely those I_para_/I_dia_ ratios corresponding to long distances. If one considers that the estimated error corresponds to the standard deviation of a Gaussian distribution, 68% of these “null PRE” values should be found between 0.9 and 1.10 (1 standard deviation), while another 15% should be found between 1.1 and 1.2. Thus the numbers of I_para_/I_dia_ ratios higher than 1.1 are well within the statistics.

The new sets of restraints measured after re-processing all spectra were used to repeat all steps of structure calculation, selection and ensemble scoring. This process yielded structures nearly identical to those of the original submission. However, the exact numbers reported from Figure 3 onwards are slightly different from the original submission. All new structures and restraints are included in an updated deposition in Dryad.

In addition, we have validated our PRE data with intra-subunit distances, as shown now in Figure 2—figure supplement 4 for fibrillarin.

The reviewers criticize that we invoke the presence of a closed conformation on the basis of I_para_/I_dia_ ratios < 0.75, while we ignore a value of 1.25. As explained above, the value of 1.25 in Figure 2 is an isolated outlier (as always in a distribution of experimental values, not all points are within one standard deviation from the average), while the values to which we refer when invoking the closed conformation comprise 6 residues in a row, with values of I_para_/I_dia_ = 0.5 in the D’-loaded complex and 0.75 in the D-loaded complex. We think that this piece of evidence, among others (see above), demonstrate that these intensity ratios are significant.

As the NMR data do not agree with SANS data, the authors involve conformational rearrangements between open and closed conformations. Clearly, in case things are not in agreement, it can always be explained by dynamics. Because of the very weak NMR evidence (2 points above), it seems a far stretch to conclude that the complex undergoing open-closed motions.

Please see the evidence for the conformational exchange now provided in Figures 3, 4 and 5 and their supplements. The ability of integrative structural biology to detect conformational exchange when data from only one technique could or would fail is exactly the point that we want to make here.

Previous point #10; Figure 2D. The figure is still hard to understand. The substrate D loaded experimental (red) and theoretical (brown) are very different. The substrate D' loaded experimental (cyan) and theoretical (blue) are also very different.

The figure has been removed. Instead of P(r) we now show the direct fit to the ^2^H-Fib SANS curves in several figures and supplements.

[Editors' note: further revisions were suggested prior to acceptance, as described below.]

First, when there are issues raised by the reviewers that seem unreasonable to the authors, this sometimes reflects the fact that the material was not explained as clearly as might be possible. One of the advantages of eLife is that space is not nearly as constrained as in some other prestigious journals. […] Perhaps if you show more PRE data this will help – and as you will hopefully be showing correlation plots assuming 1 state and that these are not as good as the intra-validation – showing more raw data would be justified.

In brief, for this revised version, we have re-analysed PRE data after reprocessing all spectra, redid all structure calculations and ensemble scoring with the new sets of PRE data (very similar to the previous ones), added new analysis and validation figures and rewritten the Results section. In addition, we no longer discuss chemical shift perturbations to support the existence of conformational exchange. The conclusions are identical to the first version of the manuscript: the fact that we now refer to a new round of structural calculations justifies the small differences in the exact numbers reported from Figure 3 onwards.